# Occurrence and source apportionment of perfluoroalkyl acids (PFAAs) in the atmosphere in China

Deming Han[1], Yingge Ma[2], Cheng Huang[2], Xufeng Zhang[1], Hao Xu[1], Yong Zhou[1], Shan Liang[1], Xiaojia Chen[1], Xiqian Huang[1], Haoxiang Liao[1], Shuang Fu[1], Xue Hu[1], Jinping Cheng[1]

[1] School of Environmental Science and Engineering, Shanghai Jiao Tong University, Shanghai 200240, China

[2] State Environmental Protection Key Laboratory of the Formation and Prevention of Urban Air Pollution Complex, Shanghai Academy of Environmental Sciences, Shanghai 200233, China

*Correspondence to*: Jinping Cheng (jpcheng@sjtu.edu.cn)

**Abstract:**

Perfluoroalkyl acids (PFAAs) are a form of toxic pollutant that can be transported across the globe and accumulated in the bodies of wildlife and humans. A nationwide geographical investigation considering atmospheric PFAAs via XAD–Passive Air Sampler was conducted in 23 different provinces/municipalities/autonomous regions in China, which provides an excellent chance to investigate their occurrences, spatial trends, and potential sources. The total atmospheric concentrations of thirteen PFAAs (n=268) were 6.19–292.57 pg/m$^3$, with an average value of 39.84±28.08 pg/m$^3$, which were higher than other urban levels but lower than point source measurements. Perfluorooctanoic acid (PFOA) was the dominant PFAAs (20.6%), followed by perfluorohexanoic acid (PFHxA), perfluorooctane sulfonate (PFOS), and perfluoroheptanoic acid (PFPeA). An increasing seasonal trend of PFAAs concentrations was shown as summer < autumn < spring < winter, which may be initiated by stagnant meteorological conditions. Spatially, the content of PFAAs displayed a declining gradient trend of central China> northern China> eastern China> northeast of China> southwest of China> northwest of China> southern China areas, and Henan contributed as the largest proportion of PFAAs. Four sources of PFAAs were identified using a positive matrix factorization (PMF) model, including PFOS–based products (26.1%), PFOA–based, and PFNA–based products (36.6%), degradation products of fluorotelomere–based products (15.5%), and an unknown source (21.8%).

## 1.Introduction

Perfluoroalkyl acids (PFAAs) are one class of ionic polyfluoroalkyl substances (PFASs), which have excellent characteristics in terms of chemical and thermal stability, high surface activity, and water and oil repulsion (Lindstrom et al., 2011;Wang et al., 2014). They are applied to a wide variety of domestic and industrial products such as textiles, oil

and liquid repellents, firefighting foam, pesticides, and food packaging materials (Xie et al., 2013;Wang et al., 2014).
PFAAs can be released to the surrounding environment during manufacturing and use of PFAAs containing products,
which are ubiquitous in the environment (e.g., in the atmosphere, water, or snow) (Dreyer et al., 2009;Wang et al.,
2017;Hu et al., 2016), in wildlife (Sedlak et al., 2017), and even in the human body (Cardenas et al., 2017;Tian et al.,
2018). PFAAs can change adult thyroid hormone levels, reduce newborn birth weight, and biomagnify up the food chain,
which can be extremely toxic to animals and humans (Hu et al., 2016;Jian et al., 2017;Baard Ingegerdsson et al., 2010).
Of the PFAAs, the long–chain (C ≥8) perfluoroalkyl carboxylic acids (PFCAs) and (C ≥7) perfluoroalkyl sulfonic acids
(PFSAs) are more toxic and bio–accumulative than their short–chain analogues (Buck et al., 2011). This especially
applies to perfluorooctanoic acid (PFOA), perfluorooctane sulfonate (PFOS) and perfluorohexane sulfonate (PFHxS), in
which PFOS and PFOA have been added to Annex B and Annex A of the Stockholm Convention in 2009 and 2019,
respectively, while PFHxS was under review by the Persistent Organic Pollutants Review Committee (Johansson et al.,
2008; UNEP Stockholm Convention, 2019).
PFAAs can originate from direct sources of products' emissions as well as indirect sources of incomplete degradation of
their precursors. It is estimated that the global historical emission quantities of C4–C14 PFCAs were 2610–21400 t in the
period of 1951–2015, of which PFOA–based and perfluorononanoic–acid (PFNA)–based products contributed the most
(Wang et al., 2014). A trend of geographical distribution of major fluorochemical manufacturing sites has shifted from
Western Europe, US, and Japan to the emerging economies in the Asia Pacific area over the past decades. This is
especially true for China, which was the world's largest industrial contributor of PFOAs (50–80 t) and PFOS–related
compounds (~1800 t) in 2009 (Xie et al., 2013). PFOA– and PFOS– based products were added to the Catalogue for the
Guidance of Industrial Structure Adjustment in China in 2011, and restricted elimination of PFOA/PFOS substances
production were conducted. With a large quantity of PFAAs and their products manufacturing and consumption, China
has become the emerging contamination hotspots in the world. In spite of several studies on atmospheric PFAAs levels
having been conducted in a few cities (Liu et al., 2015) and point sources (Yao et al., 2016a;Tian et al., 2018) in China,
due to the imbalanced urbanization and industrialization levels, there is still a lack of systemic research on atmospheric
PFAAs quantification and trends in China.
Additionally, the long range or mesoscale transport was also suggested to have a contribution to PFAAs in the air (Dreyer
et al., 2009;Cai et al., 2012a). In general, three pathways/hypotheses for the transportation of PFAAs were suggested:
transport associated with particles, degradation from precursor, and sea salts from current bursting in coastal areas. The
PFAAs precursors such as fluorotelomere alcohols (FTOHs), which can form the corresponding PFAAs through
oxidation reactions initiated by hydroxyl radicals (OH·) in the atmosphere (Thackray and Selin, 2017), are more volatile
than PFAAs and can reach remote areas via long–range transportation (Martin et al., 2006; Wang et al., 2018). Due to the
lower acid dissociation coefficient (p$K_A$), 0–3.8 for PFCAs and –3.3 for PFSAs, PFAAs are expected to be mainly
associated with aerosols in the non–volatile anionic form (Lai et al., 2018; Karásková et al., 2018). However, recent field
studies have confirmed their occurrence in gaseous phase (Cassandra et al., 2018;Ahrens et al., 2013), e.g. Fang et al.,
(2018) found the total concentrations of C2, C4–C10 PFCAs and C6 and C8 PFSAs in the gas phase were 0.076–4.0
pg/m$^3$ in the air above the Bohai and Yellow Seas, China. Investigating the transport pathways of PFAAs in nationwide
region via active air sampler (AAS) is challenging, due to their electronic power supply and high cost. Fortunately, a
number of reports showed that the XAD (a styrene–divinylbenzene copolymer) impregnated sorbent based passive air
sampler (SIP–PAS) and XAD based PAS (XAD–PAS), were proven to be an ideal alternative sampling tool for
monitoring PFAAs in a wide region. Despite several publications suggested XAD-PAS collects primarily gaseous PFAAs
in the ambient (Melymuk et al., 2014; Lai et al., 2018), current findings were not consistent. Due to the unimpeded
movements of particles into the sampler, XAD–PAS was indicated to collect a representative sample of both gas and
particle phases (Ahrens et al., 2013; Okeme et al., 2016; Karásková et al., 2018). Moreover, the dominant sorbent for
fluorinated compounds was reported as XAD resin in the XAD impregnated SIP–PAS, instead of PUF themselves
(Krogseth et al., 2013). XAD–PAS give PFASs profiles that were more closely resembled to those from AAS in
comparing with PUF–PAS, have sufficient uptake rates for the PFCAs and PFSAs to be deployed for short time duration
(Lai et al., 2018).
Given the factors mentioned above, we conducted a nationwide survey of PFAAs in China at a provincial level using a
XAD–PAS from January to December in 2017. The objective of this research was: (1) to examine the tempo–spatial
variations of PFAAs, and (2) to identify their potential affecting factors and evaluate the affecting pathways. To the best
of our knowledge, this is the first research paper analyzing both a long–term and nationwide atmospheric PFAAs data set
complemented by a comprehensive investigation in China.
**2.Material and methods**
**2.1 Chemicals and reagents**
The PFAAs standards used were Wellington Laboratories (Guelph, ON, Canada) PFAC–MXB standard materials,
including C5–C14 PFCAs analogues (Perfluoropentanoic acid (PFPeA), Perfluorohexanoic acid (PFHxA),
Perfluoroheptanoic acid (PFHpA), PFOA, Perfluorononanoic acid (PFNA), Perfluorodecanoic acid (PFDA),
Perfluoroundecanoic acid (PFUdA), Perfluorododecanoic acid (PFDoA), Perfluorotridecanoic acid (PFTrDA), and
Perfluorotetradecanoic acid (PFTeDA)), as well as C4, C6, and C8 PFSAs analogues (Perfluorobutane sulfonic acid
(PFBS), PFHxS, and PFOS). The mass–labeled $1,2-^{13}C_2$–PFHxA, $1,2,3,4-^{13}C_4$–PFOA, $1,2,3,4,5-^{13}C_5$–PFNA,
$1,2-^{13}C_2$–PFDA, $1,2-^{13}C_2$–PFUdA, $1,2-^{13}C_2$–PFDoA, $^{18}O_2$–PFHxS, and $1,2,3,4-^{13}C_4$–PFOS were used as internal
standards (ISs, MPFAC–MXA, Wellington Laboratories Inc.) in high–performance liquid chromatography (HPLC)
coupled with a tandem mass spectrometer (MS/MS). HPLC–grade reagents that were used include methanol, ethyl
acetate, ammonia acetate, acetone, methylene dichloride, n–hexane, and Milli–Q water. Detailed sources of the target
PFAAs and their ISs are listed in Table S1 in the Supplementary Materials.
**2.2 Sample collection**
Sampling campaigns were carried out at 23 different provinces/municipalities/autonomous regions in China
simultaneously from January to December 2017, of which 20 were urban sites and three were rural sites (Zhejiang,
Shanxi, and Liaoning). Urban samples typically came from urban residential areas, and the rural samples were obtained
from villages. These sampling sites were divided into seven administrative divisions: norther China (NC, n=3 sites),
southern China (SC, n=2), central China (CC, n=3), eastern China (EC, n=7), northwest of China (NW, n=3), northeast of
China (NE, n=2), and southwest of China (SW, n=3). A geographical map of the sampling sites is displayed in Figure S1,
and the detailed information on sampling sites such as elevation , meteorological parameters, local resident population
and gross domestic product were listed in Table S2 and Figure S1.
Samples were collected with Amberlite XAD–2 resin using XAD–PAS, which have been successfully monitored PFCAs
(C4–C16) and PFSAs (C4–C10) in the atmosphere (Krogseth et al., 2013;Armitage et al., 2013). Briefly, the mesh
cylinder (L.× I.D.: 10 cm × 2 cm) was prebaked at 450°C for 3 h, filled with ~10 g XAD–2 resin, and capped with an
aluminum cap. The particle size of XAD-2 is ~20-60 mesh, with water content of 20%-45%, its specific surface area
$\geq 430$ $m^2$/g, and the reference adsorption capacity $\geq 35$ mg/g. We should keep in mind that the unimpeded movement of
particle bound PFAAs would be captured during sampling using XAD-PAS, which cannot differentiate PFAAs between
gas and particle phases. Despite some research suggest the sampling efficiency of gas and particle phase PFAAs were
similar (Karásková et al., 2018). In the present study, the reported PFAAs sampled by XAD-PAS represent a combination
of gaseous and particulate PFAAs concentration. The sampling program for each sample lasted approximately a month
(30 days), and the error of the sampling time was controlled within 3 d. At the end of each deployment period, the
atmosphere samples were retrieved, resealed in their original solvent–cleaned aluminum tins at the sampling location,
and transported by express post to Shanghai Jiao Tong University. On receipt, they were stored and frozen (–20 °C) until
extraction.
The sampling rate of XAD–PAS is a crucial factor to derive the chemical concentrations accumulated in the XAD resin.
Ahrens et al. (2013) found that sampling rate of PFCAs and PFASs ranged 1.80–5.50 $m^3$/d with XAD impregnated
sorbent, and the sampling rate increased as the carbon chain adding, while Karásková et al. (2018) suggested that the
sampling rate of XAD–PAS of 0.21–15.00 $m^3$/d for PFAAs. The loss of depuration compounds could be used to calculate
the sampling rate, assessing the impacts from meteorological factors like temperature and wind speed. According to
Ahrens et al. (2013) the 1,2,3,4–$^{13}C_4$–PFOA was used to calculate the sampling rates of PFAAs at Shanghai sampling site
(Shanghai Jiao Tong University) in the present study, by assessing 1,2,3,4–$^{13}C_4$–PFOA abundance loss. The specific
description of the sampling rate calculation in this study is shown in Section S1 in the Supplementary Materials.
**2.3 Sample preparation and instrument analysis**
The sample preparation and analysis were according to the method described by previous researches (Liu et al.,
2015;Tian et al., 2018). The MPFAC–MXA ISs mixture surrogates (10 ng) were added to each spiked sample prior to
extraction. This was done to account for the loss of substances from the samples associated with instrument instability
caused by the changes in laboratory environmental conditions. The XAD resin samples were Soxhlet–extracted for 24 h
using a Soxhlet extraction system, with n–hexane: acetone (1:1, V:V) as a solvent in a 300 mL polypropylene (PP) bottle,
following extracted with methanol for 4 h. These two extracts were combined and reduced to ~5 mL via a rotary
evaporator (RE–52AA, Yarong Biochemical Instrument Inc., Shanghai, China) at a temperature below 35 °C, and then
transferred to a 10 mL PP tube for centrifugation (10 min, 8,000 rpm). The supernatant was transferred to another PP tube,
filtered three times through a 0.22 μm nylon filter, with an addition of 1 mL methanol each time. The extracts were
further condensed under a gentle stream of nitrogen (99.999%, Shanghai Liquid Gas Cor.) at 35 °C to a final 200 μL for
instrument analysis.
The separation and detection of PFAAs were performed using a HPLC system (Thermo Ultra 3000$^+$, Thermo Scientific,
USA) coupled with a triple quadrupole negative electrospray ionization MS/MS (Thermo API 3000, Thermo Scientific,
USA). An Agilent Eclipse XDB C18 (3.5 μm, 2.1 mm, 150 mm) was used to separate the desorbed substances. The
column temperature was set to 40 °C, and the flow rate was 0.30 mL/min. The injection volume was 20 μL. The gradient
elution program of the mobile phase A (5 mmol/L aqueous ammonium acetate) and B (methanol) was 80% A + 20% B at
the start, 5% A + 95% B at 8 min, 100% a at 13 min, 80% A + 20% B at 14 min, and was maintained for 6 min. The
MS/MS was operated in a negative ion scan and multiple reaction monitoring (MRM) mode, and the electrospray voltage
was set to 4500 V. The ion source temperature was 450 °C. The flow rates of the atomization gas and air curtain gas was
10 and 9 L/min, respectively. Species identification was achieved by comparing the mass spectra and retention time of
the chromatographic peaks with the corresponding authentic standards.

**2.4 Quality assurance and quality control**

To avoid exogenous contamination, the XAD–2 resin was precleaned using a Soxhlet extraction system with acetone and
petroleum ether at extraction times of 24 h and 4 h, respectively. The extracted XAD resin was dried under a vacuum
desiccator, wrapped in an aluminum foil and zip–lock bags, and stored at –20 °C to avoid contamination. All laboratory
vessels were PP, and these vessels were washed with ultrapure water and methanol three times, respectively.
For quantification, six–point calibration curves of PFAAs were constructed by adopting different calibration solutions
with values of 1, 3, 6, 15, 30, and 60 ng/mL. The same concentration for the internal calibration (10 ng/mL) was used for
each level of the calibration solution. Recovery standards were added to each of the samples to monitor procedural
performance, and the mean spiked PFAAs recoveries ranged from 81%±25% to 108%±22%. All the analyzed PFAAs
were normalized against the recovery of the corresponding mass–labeled ISs. Field blanks were prepared at all sampling
sites, transported, and analyzed in the same way as the samples. Laboratory blanks were obtained by taking amounts of
solvent via extraction, cleanup, and analysis. A total of 8 field blanks and 26 laboratory blanks were analyzed, with
individual blank values of BDL (below detection limit)–1.12 $pg/m^3$ and BDL–1.29 $pg/m^3$, respectively. All the results
were corrected according to the blank and recovery results. The method detection limit (MDL) was derived from three
times standard deviation of the field blank values. The limit of detection (LOD) and the limit of quantification (LOQ)
were determined as a signal–to–noise ratio of 3:1 and 10:1, respectively (Rauert et al., 2018;Liu et al., 2015). To convert
MDLs, LODs and LOQs values to $pg/m^3$, the mean volume of sampling air ($m^3$) was applied. For the analytes that were
not detected or were below the LOQs in field blanks, MDLs were derived directly from three times the corresponding
LODs. More detailed information on the individual compounds of PFASs on MDL, LOD, LOQ, the recovery values, and
blank values are listed in Table S3.

**2.5 Statistical and geostatistical analysis**

Statistical analyses were carried out by SPSS Statistics 22 (IBM Inc. US), and the values of 1/2 MDL were used to
replace these measured results of BDL. The statistics figures were depicted using technical software of SigmaPlot 14.0
(Systat Software, US). And the geographical variations of atmospheric PFAAs were analyzed with ArcGIS 10.4 (ESRI,
US). The Hybrid Single-Particle Lagrangian Integrated Trajectory (Hysplit) back trajectory model (NOAA, US) was used
to study the long range transport of air masses in the sampling locations (Zhen et al., 2014). Positive matrix factorization
(PMF) is considered an advanced algorithm among various receptor models, which has been successfully applied for
source identification of environmental pollutants (Han et al., 2018;Han et al., 2019) . PMF (5.0, US EPA) was adopted to
cluster the PFAAs with similar behaviors to identify potential sources, and a more detailed description of PMF can be
seen in Section S2.

**3.Results and discussion**

**3.1 Abundances and compositions**

The descriptive statistics of all targeted atmosphere PFAAs (n=268) are presented in Table 1 and Table S4. The total
concentrations of $\Sigma_{13}$ PFAAs analogues varied between 6.19 and 292.57 pg/m$^3$, with an average value of 39.84±28.08
pg/m$^3$. The commonly concerned PFCAs analogues (C5–C14) occupied 79.6% of the total PFAAs, at a level of
4.50–247.23 pg/m$^3$, whereas the PFSAs concentrations were 1.04–42.61 pg/m$^3$. The long–chain PFCAs concentrations
were 17.96±13.71 pg/m$^3$, which were significantly higher than the short–chain concentrations (13.74±12.19 pg/m$^3$)
(p<0.05). Similarly, a recent PFAAs measurement conducted in the landfill atmosphere in Tianjin, China (Tian et al.,
2018), found the long chain PFCAs were much higher than the short species. Specifically, PFOA was the dominant
PFAAs (accounting 20.6%), and was detected in all atmospheric samples with an average value of 8.19±8.03 pg/m$^3$. This
phenomenon could occur since PFOA is widely used in the manufacturing of polytetrafluoroethylene (PTFE),
perfluorinated ethylene propolymer (FEP), and perfluoroalkoxy polymers (PFA) (Wang et al., 2014). The domestic
demand for and the industrial production of PFOA–based products have been increasing in China since the late 1990s
(Wang et al., 2014), and direct emissions of FOSA–based products may contribute to the relative high level of PFOA.
Meanwhile, one major variation of PFOA precursor, 8:2 FTOH, was reported to rank as the highest concentration among
neural PFASs in air of China (De Silva, 2004;Martin et al., 2006). Among PFAAs' composition profile, it was followed
by PFHxA, PFOS, and PFPeA, with mean concentrations of 5.36, 5.20, and 4.95 pg/m$^3$, respectively. The detection
frequencies of PFCAs decreased gradually as the carbon chain length increased – for instance, the PFPeA and PFTrDA
were detected in 84.8% and 37.3%, respectively.
Compared with other gaseous PFAAs measurements, Liu et al. (2015) reported that PFAAs in the urban atmosphere
sampled with XAD–containing sorbent in Shenzhen city in China was 15±8.8 pg/m$^3$, which contributed to nearly half of
this study. Wong et al. (2018) found that a much lower PFAAs levels in the remote Arctic area than this study, with mean
value of 1.95 pg/m$^3$. This study found generally higher PFAAs abundances compared to measurement in Canada
(Gewurtz et al., 2013), which may be attributed to the relative high abundance of industrial and domestic emissions in
China. However, the PFAAs concentrations in urban/rural areas in this study were far lower than the measurements at

point sources, for example, landfill atmosphere (Tian et al., 2018) (360–820 pg/m$^3$) and fluorochemical manufacturing facility (Chen et al., 2018) (4900±4200 pg/m$^3$), suggesting that PFAAs were susceptible to being affected by local source emissions. Although there existed inherent differences of PFAAs levels between regions, the impacts from differences in sampling techniques and sorbents between XAD-PAS and SIP-PAS could not be neglected. As indicated by previous researches, XAD has much higher sorptive capacity of PFASs than PUF, wind speed and temperature displayed different degrees of impact on their sampling capacity among different regions. Additionally, UV radiation has the potential to degrade PFAAs due to $O_3$, OH·, and other atmospheric oxidants during sampling.

**Table 1.** Comparison of PFAAs levels in the present research with measurements in other areas (pg/m$^3$)

| Sampling sites | Duration | Sampling location | Sampler type [a] | PFAAs [b] | PFCAs [c] | Reference |
|---|---|---|---|---|---|---|
| 23 provinces in China | 2017.1–12 | Urban and rural areas | XAD–PAS | 6.19–292.57; 39.84±28.08 | 4.50–247.23; 31.69±23.88; C5–C14 | This study |
| Shenzhen, China | 2011.9–11 | Urban area | SIP–PAS | 3.4–34; 15±8.8 | 11.59±8.74; C4–C12 | (Liu et al., 2015) |
| Fuxin, China | 2016.9–10 | Fluorochemical manufacturing facilities | SIP–PAS | 4900±4200 | 4900±4200; C4–C12 | (Chen et al., 2018) |
| Tianjin, China | 2013 | Waste water treatment plant | SIP–PAS | 87.9–227; 123 | 87.9–227; 123; C6–C12 | (Yao et al., 2016a) |
| Tianjin, China | 2016.5–6 | Landfill | SIP–PAS | 280–820 | 280–820; C4–C12 | (Tian et al., 2018) |
| Canada | 2006–2011 | Remote and urban areas | SIP–PAS | 0.014–0.44 | 0.014–0.44; C8–C12 | (Gewurtz et al., 2013) |
| Alert, Arctic | 2006.8–2015.2 | Remote area | SIP–PAS | 1.95 | 1.95; C4–C8 | (Wong et al., 2018) |
| Toronto, Canada | 2010.3–10 | Semi–urban site | SIP–PAS | 11.24±7.95 | 11.24±7.95; C4–C18 | (Ahrens et al., 2013) |
| Brno, Czech Republic | 2013.4-9 | Suburban background site | XAD–PAS | 30–153 | 26–147.6; C4–C14 | (Karásková et al., 2018) |

[a]: SIP-PAS represent XAD impregnated PUF sorbent based PAS, which is composed of PUF, mashed XAD-4, and PUF;

[b]: represent the total concentration ranges of PFCAs and PFSAs; mean concentrations of the total PFCAs and PFSAs;

[c]: represent concentration range; mean value; carbon length of PFCAs.

## 3.2 Temporal variations

Monthly and seasonal variations of the mean PFAAs concentrations are depicted in Figure 1. In general, an increasing seasonal mean of PFAAs concentrations from 23 sampling sites existed for summer (31.35 pg/m$^3$) < autumn (35.63 pg/m$^3$) < spring (42.40 pg/m$^3$) < winter (52.83 pg/m$^3$). The winter maxima abundance of PFAAs could be attribute to the stagnant atmospheric conditions, in which atmospheric contaminants were trapped in the air with a weak diluting effect. XAD–PAS showed similar efficiency of capturing gas and particle phases PFASs, while the unimpeded particle gathering efficiency is challenging to quantify. In addition, despite the increase in atmospheric oxidation of precursors in summer may lead to PFCAs rise (Li et al., 2011;Yao et al., 2016a), the abundant rainfall would enhance their scavenging activities (Table S5), ultimately leading to the relatively low concentrations of PFAAs in the summer. Specifically, the PFAAs showed much higher concentrations in spring than other seasons in Shanghai, which was different from Tianjin and Xinjiang (Figure S2). An extreme high level of PFAAs of 135.51 pg/m$^3$ was occurred in November in Beijing, which was 2–4.5 times higher than in other month, indicating the potential point source of PFAAs contamination in this site. In fact, numerous fluoride related products manufacturers were distributed in EC, NC (including Beijing) and CC areas, see detail in Figure S3. As gaseous PFAAs measurements were majorly reported at a relative short time (several weeks to several months), it is somewhat difficult to compare their temporal trends.

Interestingly, the evolution of PFAAs showed a dramatic monthly variation, and the monthly mean levels varied from 25.92 to 60.57 pg/m$^3$, with the lowest and the highest abundances being present in September and December, respectively. For the specific composition profile of PFAAs, the average concentrations of PFOA, PFHxA, PFPeA, and PFOS were 10.36, 8.42, 6.55, and 6.44 pg/m$^3$ in winter, respectively, which were nearly two times higher than in the summer. The seasonal variation trend of PFOS was summer < spring ≈ autumn < winter, while PFNA appeared to show winter maxima with concentrations 4 and 3 times higher than in the summer and spring, respectively. However, Wong et al. (2018) reported that PFBS showed the maximal value in winter but found no consistent seasonality for PFOS in the Arctic area. The differences may be explained as the PFAAs in air in the remote Arctic area were originated from long–range transport and volatilization from snow or sea, but not affected by local direct anthropogenic emission.

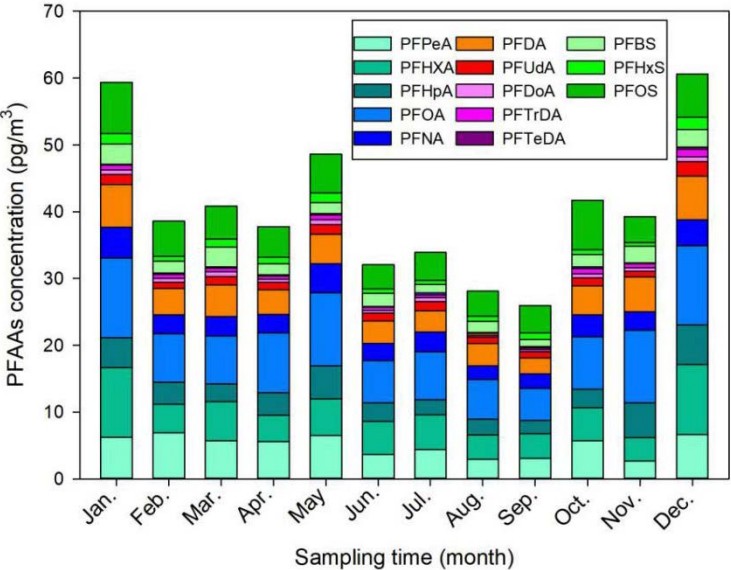


238        **Fig. 1.** Monthly mean concentrations of PFAAs in China from January to December 2017


**3.3 Geographical distributions**
Due to the stark differences in topography and socioeconomic development of Chinese provinces, municipalities, or
autonomous regions, as well as the enormous differences in industrialization and emissions, PFAAs showed significantly
different distribution patterns in China (Figure 2). Overall, the predominant declining gradient of PFAAs' contents was
CC> NC> EC> NE> SW> NW> SC areas in China, which was similar to previous research that the outdoor
dust–bounded PFAAs were relatively enriched in the eastern part of mainland China (Yao et al., 2016b). This trend was
not surprising since numerous PFAAs related photoelectric industries, chemical industries, and mechanical industries are
dispersed across CC, EC and NC areas, e.g., Shanghai, Zhejiang, Fujian, Henan, and Jiangsu. As expected, the western
mountain and highland areas, e.g., Xinjiang and Yunnan (20.88 pg/m$^3$), with relatively low population densities and high
latitudes, displayed significantly lower PFAAs concentrations. It was reported that high orographic conditions have a
cold trapping effect on atmospheric PFASs, the transportation of PFAAs involving particles or not should be dramatically
reduced (Konstantinos et al., 2010;Yao et al., 2016a). Given that altitudes increase gradually from several meters in EC,
NC and SC coastal areas to nearly 2,000 meters in SW and NW highland regions in China, the high altitude blocking
effect for atmospheric PFAAs transportation should not be neglected.
The annual average concentrations of PFAAs at the provincial level ranged from 12.38 pg/m$^3$ in Xinjiang to 90.88 pg/m$^3$
in Henan, and the composition patterns varied widely. Henan contributed the largest proportion of PFAAs in China, and
showed the highest PFOA level (19.07 pg/m$^3$), which is a typical, heavily–industrialized province characterized by textile

treatments, metal plating, and firefighting foam manufacturing, and a large amount of PFAAs emulsifier fluoropolymers were used in industrial production. Special attention should be paid to Zhejiang, the level of which (61.68 pg/m$^3$) ranked second in PFAAs abundances in spite of its sampling site being located in a village. As well as this, several painting–packaging plants, mechanical plants, and electrical equipment manufacturers were dispersed around this sampling site (see Figure S4), which would contribute to the PFAAs variations in this site. In fact, the GDP of Zhejiang ranked fourth in China, specializing in mechanical manufacture, textiles, and chemical industry. Moreover, the top six sites with abundant of PFAAs were located in the most economically–developed and populated areas (the Yangtze River Delta area, the Circum–Bohai Sea Region), and in the rapidly–developing regions (Henan, Sichuan) in China. In line with this result, a sampling campaign conducted in Asia, including 18 sites in China, found very high levels of PFAAs precursors (8:2 FTOH, 10:2 FTOH) existed in Beijing, Tianjin, and Zhejiang (Li et al., 2011). But meanwhile we should keep in mind that the production of PFCAs in the atmosphere from gaseous precursors degradation may be impaired in urban areas, due to the high abundance of NOx compete for OH· radicals.

Furthermore, PFOA concentrations were apparently high in Henan, Zhejiang, Beijing, Tianjin, and Hubei, where mean values ranged of 11.65–19.14 pg/m$^3$ compared with in other provinces (2.93–8.54 pg/m$^3$). PFOA and PFOA–related products have not been banned for use in various industrial and domestic applications (Konstantinos et al., 2010;Wang et al., 2014), which were manufactured extensively in EC and NC areas and were used widely. However, the highest concentration of PFOS was found in Zhejiang (14.13 pg/m$^3$), which may be affected by local manufacturing of PFOS based products, e.g. leather, paper and metal plating. It was followed by Beijing (8.98 pg/m$^3$) and Fujian (9.09 pg/m$^3$), while Xinjiang and Yunnan shared the lowest levels (1.20–3.57 pg/m$^3$). This spatial variation patterns of PFOS in the present study, matched well with a previous national survey that found most PFOS and its derivative facilities in China are suited in EC, CC and NC areas, with emission density ranged from 1–500 g/(km$^2$·a) (Konstantinos et al., 2010;Wang et al., 2014).

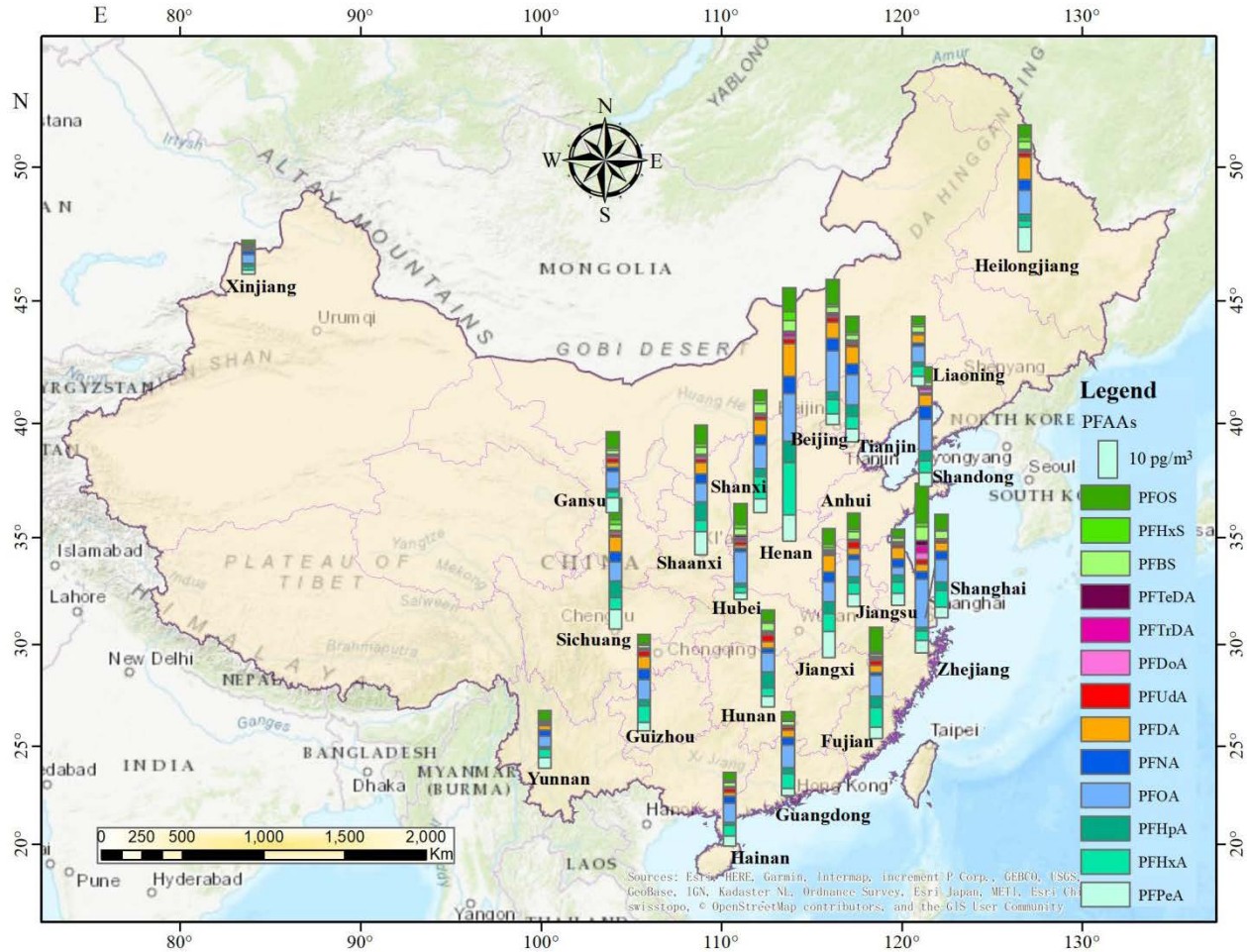

**Fig. 2.** The spatial distributions of PFAAs in China (annual average of PFAAs, created by ArcGIS 10.4).

**3.4 Geographical distributions transport pathway**
The PFAAs variations in the atmosphere depended on their local source emissions as well as regional atmosphere
transportation. In order to give readers a direct impression of factors affecting the geographical variations of PFAAs in
China, here we analyzed PFAAs variations along three pathway transects and one coastal line to determine how PFAAs
distribute spatially.
As shown in Figure 3a, PFAAs concentrations were enriched in southeastern areas (40.58–47.17 pg/m$^3$) at low altitudes
(2–30 m), but relatively low abundances (12.31–29.44 pg/m$^3$) existed in the northwestern part of China (397–1,517 m in
altitude). As discussed above, the EC areas (e.g. Fujian) were the most intensively industrialized regions, direct emissions
from PFAAs manufacturing processes would enhance their atmospheric abundances. However, high altitudes existed in
NW areas would have a blocking effect to the transportation of PFAAs from eastern polluted areas.
In terms of the SW–NE transect (Figure 3b), Yunnan and Liaoning showed much lower PFAAs concentrations (20.88 and
24.99 pg/m³) than other areas (44.76–52.58 pg/m³). Notably, a steady increasing trend of PFAAs concentrations existed
across the W–E transect (Figure 3c), which escalated from 20.88 pg/m³ in Yunnan to 61.68 pg/m³ in Zhejiang. The
composition profiles of PFAAs along this transect differed from each other; for instance, PFOA occupied 28.5% of the
total PFAAs in Zhejiang, while it only accounted for 15.6%–21.8% in other areas. Note that PFAAs released from point
sources would be eliminated by deposition, degradation, or dilution during transportation in the atmosphere, e.g., PFOA
could decrease by ~90% within 5 km of its point source (Chen et al., 2018). However, the long range transport of PFAAs
bounded with particles also have been explored in previous research (Pickard et al., 2018). As illustrated in Figure S5, the
48 hours back trajectories were generally associated with air masses originating from the surrounding areas of the
sampling locations, the trajectories which overlapped with urban areas in Zhejiang, Jiangxi and Shanghai, which
confirmed that the air mass origins was a driving factor for PFAAs variation.
Interestingly, with the exclusion of the site directly affected by surrounding sources in Zhejiang, PFAAs were rather
uniformly distributed among the coastal areas, with concentrations ranging from 24.92–45.76 pg/m³ (Figure 3d).
Excluded industrial and domestic emissions as well as secondary formation, the PFAAs containing sea spray aerosols
could contribute the variations of PFAAs in coastal atmosphere (Cai et al., 2012b; Pickard et al., 2018).

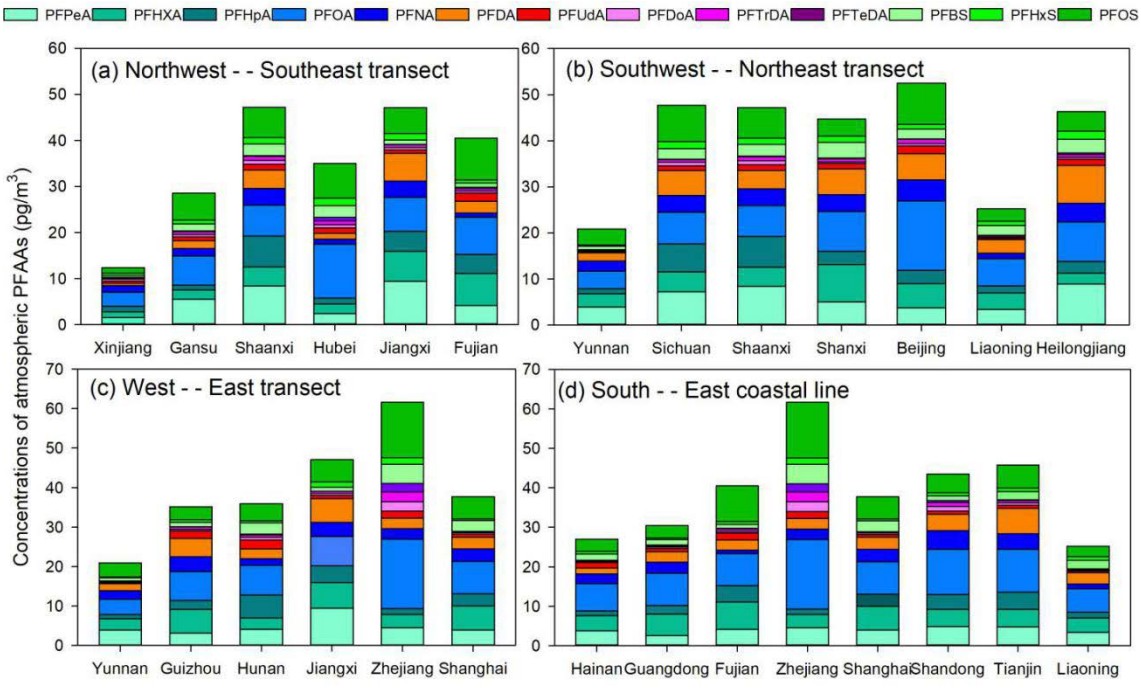

**Fig. 3.** Transects of PFAAs concentrations across three different directions and one coastal line

**3.5 Source identification**
Understanding the sources of PFAAs and their corresponding importance would enable elucidation of the levels of
PFAAs in the environment. As discussed above, the observations from tempo–spatial variations of PFAAs suggest that
several factors may have a combined effect on the variations of PFAAs. Hence, a PMF model was adopted to extract the
potential factors affecting PFAAs variations, and four sources were extracted in this study (see Figure 4).
High percentages (~90.0%) of PFPeA and PFBS were found in factor 1, and were moderately loaded with PFOS (62.6%).
Three major types of PFOS–related chemicals; namely PFOS salts, PFOS substances and PFOS polymers, are used in
industrial products in China (Xie et al., 2013). PFOS salts are usually used in metal plating, firefighting foams, and
pesticides, while PFOS substances are adopted in paper treatment and the semiconductor industry. PFOS polymers are
employed for textile and leather treatment. These PFOS–related products would lead to direct emissions of PFOS during
their industrial and domestic activities. PFPeA and PFBS are the main substitutes for long–chain PFAAs in China, which
would release as impurities or by–products when manufacturing PFOS–based products (Liu et al., 2017). Hence, this
factor was regarded as the direct source of PFOS–based products. This was consistent with the spatial observations that
high PFOS concentrations were shown in Zhejiang, Fujian, Guangdong, and Shanghai, where manufacturing facilities are
distributed.
Factor 2 was characterized by PFHxA, PFOA, PFNA, and PFDA, each representing over 60.0% of their explained
variations. Their rather strong positive correlations (r=0.54–0.84, p<0.01) suggested that they may have originated from a
similar source (Table S6). PFOA was considered as the marker for the emulsification of plastics, rubber products, flame
retardants for textiles, paper surface treatments, fire foams and PTFE emulsifiers (Liu et al., 2015;Konstantinos et al.,
2010). It has been reported that there was an increase in PFCAs emissions at the manufacturing sites of PFOA–based
products in China between 2002 and 2012 due to a rapid increase in domestic demand and production of PFOA–related
products (Wang et al., 2014). PFNA and its derivatives have similar physicochemical properties to PFOA and its
derivatives, and both can be emitted through exhaust gases. The PFNA–based production was found to be related to
polyvinylidene fluoride (PVDF) production, and it has been suggested that PVDF production increased in China after
2008 (Wang et al., 2014). Therefore, factor 2 represents direct sources of PFOA–based and PFNA–based products.
The compositions of factor 3 were characterized by a high loading of PFHpA and PFHxS, with loading factor values of
84.9% and 81.7%, respectively. The historical production and uses of PFHpA and its derivatives remain unidentified.
Factor with PFHxS alone did not indicate a specific source, so this factor may be classified as an unknown source, which
may be affected by atmosphere air mass transport, sea aerosol bursting and/or other origins.
The final factor was dominated by PFUdA, PFDoA, PFTrDA, and PFTeDA, with loading factor values larger than 80%.
These long–chain PFAAs (C11–C14) analogues have been interpreted as degradation products of fluorotelomer–based
products in previous research (Liu et al., 2017;Wang et al., 2014; Thackray and Selin, 2017). Based on the life–cycle
usage and release from fluorotelomer and other fluorinated products, the global cumulative estimation of PFUdA,
PFDoA, PFTrDA, and PFTeDA from quantified sources was estimated to be 9–230 tons in the period of 2003–2015, and
projected to be between 0–84 tons between 2016–2030 (Wang et al., 2014). It was reported that the manufacturing of
fluorotelomer–based substances would increase in China. In addition, these four analogues showed apparent positive
correlations to each other (r =0.59–0.79, p<0.01). Thus, this factor was explained as the degradation products of
fluorotelomer–based products, which could be proven by their higher abundances caused by an enhanced atmospheric
oxidation ability in the summer than other seasons.
Direct emission sources, including PFOS–based products, PFOA–based products, and PFNA–based products were
estimated to represent 62.7% of the total PFAAs sources. Indirect sources of degradation products of
fluorotelomer–based products played a minor role, contributing 15.5%, and there are 21.8% of variances that could still
not be explained and need further detailed investigation. This source apportionment result was similar to one recent piece
of research that found that industrial PFOA emissions were the major sources of atmospheric PFAAs in Shenzhen, China
(Liu et al., 2015), and the long–distance transportation of pollutants also made a contribution.

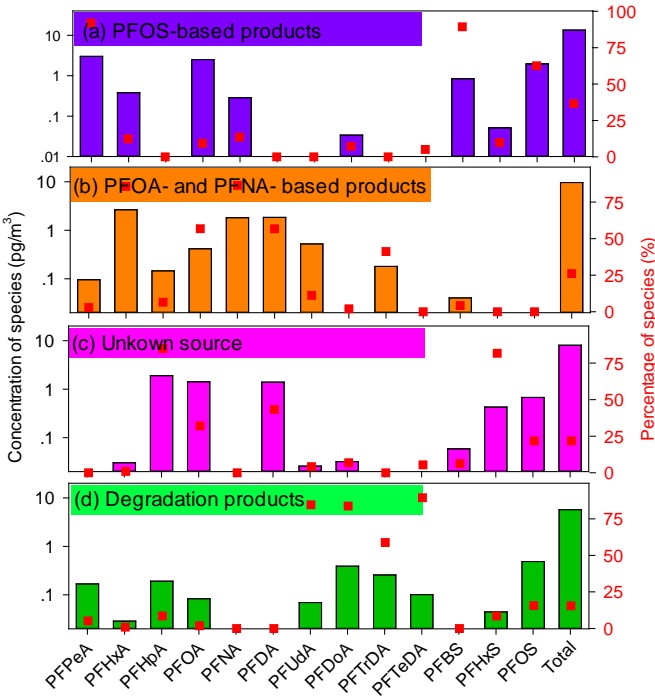


**Fig. 4.** Factor profiles of PFAAs extracted by the PMF model
**4. Conclusion**
In the present study, PFAAs were ubiquitously detected in the atmosphere across China over the length of a year. Results

indicated that the measured PFAAs in the present study were several times to several magnitudes higher than the levels conducted in most other urban locations, while far lower than the measurements implemented at point sources. In which, the C5–C14 PFCAs analogues occupied 79.6% of the total PFAAs variations, PFOA, PFHxA and PFOS ranked the top three species. Additionally, much higher abundances of PFAAs existed in winter compared with in summer. In terms of spatial distribution, the PFAAs concentrations were higher in central and eastern China, where dense residential and industrial manufacturing facilities were distributed. Correlation analysis, Hysplit backward trajectories, and PMF receptor model, have combined to suggest that the direct sources of PFOS–based, PFOA–based, and PFNA–based products made a predominant contribution to variations in PFAAs, while indirect degradation played a minor role.

**Acknowledgements**

This study was financially supported by National Key Research & Development Plan (2016YFC0200104), National Natural Science Foundation of China (No. 21577090 and No. 21777094), and China Postdoctoral Innovative Talent Support Project (BX20190169). We thank Lei Ye (Xi'an University of Architecture and Technology), Fengxia Wang (Hainan University), Linrui Jia (Beijing Normal University), Songfeng Chu (Tongji University), and other 18 volunteers, for coordinating the sampling process and for their valuable contribution to field measurement. We appreciate senior engineer Xiaofang Hu (Instrumental Analysis Center, SESE, Shanghai Jiao Tong University) for her assistance in experiment analysis.

**Appendix A: Supplementary material**

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
