# Peer review of "Occurrence and source apportionment of perfluoroalkyl acids (PFAAs) in the atmosphere in China"

_Atmospheric Chemistry and Physics, 2019_

## Referee Comment (RC1) · Anonymous Referee #1 · 12 Aug 2019

Major comment: It is very unlikely that PFSAs occur in the gas phase. Recent studies have shown that PFSAs have been measured in passive samplers since passive samplers are also collecting particles. It is really important to clarify in this manuscript that PFSAs are typically particle-bound and the measured concentrations of PFSAs are most likely due to the collection of particles using the passive samplers. Make also clear that the measured concentration is NOT the gas phase but rather "air concentrations" with mainly gas phase and partly particulate phase. This has to be clarified at all places in the manuscript, figures and SI before a publication can be considered.

General comments: - Use two significant numbers - Table 1: o It should be "HV-AAS" for Toronto, Canada - Figure 3 o Change to "Northwest" - Figure 4 o Describe the four factors in the figure caption

[Figure]

Specific comments: Line 13: Add total number of samples, number of sampling locations and which sampling method was used Line 13: indicate if PFAAs were measured in the gas or particulate phase Line 18: Specify which location are these "areas" Line 24: Change to "ionizable" Lines 187-189: Clarify that this is the average of x numbers of sampling locations in China Lines 216-218: Indicate, how many sites were included from each area "(n = . . .)"

————————————————————

---

## Short Comment (SC1) · 16 Aug 2019

A national scale passive air sampling campaign was carried out in China, and 11 perfluoroalkyl acids compounds were determined in air samples during a whole year. The authors discussed concentration profiles, distributions and potential sources. The manuscript has the potential to add to the available body of evidence. I believe that the data are reliable and useful. In general, I recommend that the manuscript be accepted pending some minor revisions as outlined below. 1. There should be a space between numbers and units, like line 96 and line 130, -20°C. 2. Line 29, the authors listed the environment like atmosphere, water, or snow, or in wildlife and even in the human body, however, the references cited seemed not match. 3. Line 32, the long-chain perfluoroalkyl carboxylic acids should be defined as C $\geq$ 8. 4. Line 74, the PFCAs analogues

abbreviations listed in brackets should be given the full name, because some of them occurred at the first time. 5. Why did not the authors collect all the samples from urban area? 6. Please give the information on Amberlite XAD-2 resin. 7. If the MDL was derived from three times SD of the field blank values, the authors should give the information about the field blanks and laboratory blanks. Which compounds were detected in those blanks? And in what level? 8. Did the authors use the matrix spike? Is there any matrix effect in passive air samples?

---

## Short Comment (SC2) · 5 Sep 2019

A list of responses for comments from editors and reviewers

Dear Editors and Reviewers:

Thank you for your letter and for the reviewers' comments concerning our manuscript entitled "Occurrence and source apportionment of perfluoroalkyl acids (PFAAs) in the atmosphere in China" (Ref: acp–2019–676). These comments are valuable and very helpful for revising and improving our paper, as well as the important guiding significance to our researches. We have studied comments carefully and have made correction, the correction in the manuscript was marked–up with blue colour and underline (e.g. Revised Manuscript) which we hope meet with approval. The main corrections in

the paper and the responds to the reviewer's comments are as flowing: Responds to the editors' and reviewers' comments:
A national scale passive air sampling campaign was carried out in China, and 11 perfluoroalkyl acids compounds were determined in air samples during a whole year. The authors discussed concentration profiles, distributions and potential sources. The manuscript has the potential to add to the available body of evidence. I believe that the data are reliable and useful. In general, I recommend that the manuscript be accepted pending some minor revisions as outlined below. Response: Thank you for reviewer's appraisal of our manuscript. We appreciate reviewer's valuable comments for improving the manuscript.

Minor revisions: Query 1. There should be a space between numbers and units, like line 96 and line 130, –20âŮęC. Response: Thanks for reviewer's suggestion. The format of number and units in lines 96 and 130 were revised in the revised manuscript, and all other were checked and revised throughout the revised manuscript.

Query 2. Line 29, the authors listed the environment like atmosphere, water, or snow, or in wildlife and even in the human body, however, the references cited seemed not match. Response: Considering reviewer's suggestion, the references cited were revised. It was changed as "PFAAs can be released to the surrounding environment during manufacturing and use of PFAAs containing products, which are ubiquitous in the environment (e.g., in the atmosphere, water, or snow) (Dreyer et al., 2009;Hu et al., 2016;Wang et al., 2017), in wildlife (Sedlak et al., 2017), and even in the human body (Cardenas et al., 2017;Tian et al., 2018)." in lines 30–33 in the revised manuscript.

Query 3. Line 32, the long–chain perfluoroalkyl carboxylic acids should be defined as

C $\geq$ 8. Response: To date, there are at least two classification for PFAAs, one for long–chain of C $\geq$ 8 and short–chain of C $\leq$ 7 (Dreyer et al. 2009; Liu et al. 2015), while the other for long–chain of C $\geq$ 7 and short–chain of C $\leq$ 6 (Liu et al. 2015; Jin et al. 2015; Tian et al. 2018). In the present study, to compare with the PFAAs variations which conducted in China recently reported, the long–chain of PFAAs was selected as C $\geq$ 7. Hence, this query has not been revised in the revised manuscript.

Query 4. Line 74, the PFCAs analogues abbreviations listed in brackets should be given the full name, because some of them occurred at the first time. Response: All of full names and abbreviations of the 13 PFAAs analogues, were listed in Table S1 in the Supporting Materials. Considering reviewer's suggestion, the description of "The PFAAs standards used were Wellington Laboratories (Guelph, ON, Canada) PFAC–MXB standard materials, including C5–C14 PFCAs analogues (PF-PeA, PFHxA, PFHpA, PFOA, PFNA, PFDA, PFUdA, PFDoA, PFTrDA, and PFTeDA), as well as C4, C6, and C8 PFSAs analogues (PFBS, PFHxS, and PFOS)." was changed to "The PFAAs standards used were Wellington Laboratories (Guelph, ON, Canada) PFAC–MXB standard materials, including C5–C14 PFCAs analogues (Perfluoropentanoic acid (PFPeA), Perfluorohexanoic acid (PFHxA), Perfluoroheptanoic acid (PFHpA), PFOA, Perfluorononanoic acid (PFNA), Perfluorodecanoic acid (PFDA), Perfluoroundecanoic acid (PFUdA), Perfluorododecanoic acid (PFDoA), Perfluorotridecanoic acid (PFTrDA), and Perfluorotetradecanoic acid (PFTeDA)), as well as C4, C6, and C8 PFSAs analogues (Perfluorobutane sulfonic acid (PFBS), PFHxS, and PFOS)." in lines 76–81 in the revised manuscript.

Query 5. Why did not the authors collect all the samples from urban area? Response: This research is a part of a large study aimed to investigate the occurrence and regional transportation of new emerging pollutants in China, in which one crucial pollutant was PFAAs. This investigation was implemented by Shanghai Jiao Tong University and Shanghai Academic of Environmental Science, and the sampling sites were selected based on the comprehensive effects of sampling geographical location, availability of volunteers, convenience of exchanging sorbent of XAD–PAS, and some other factors. Therefore, 20 urban sampling sites and 3 rural sampling sites were selected in this investigation ultimately.

Query 6. Please give the information on Amberlite XAD–2 resin. Response: Considering reviewer's suggestion, detailed information on XAD–2 was added in lines 99–100 in the revised manuscript, as "The particle size of XAD–2 is ∼20–60 mesh, with water content of 20%–45%, its specific surface area ≥430 m2/g, and the reference adsorption capacity ≥35 mg/g."

Query 7. If the MDL was derived from three times SD of the field blank values, the authors should give the information about the field blanks and laboratory blanks. Which compounds were detected in those blanks? And in what level? Response: According to reviewer's suggestion, more information about filed blanks and laboratory blanks was added in the revised manuscript and supporting materials. For instance, the description of "A total of 8 field blanks and 26 laboratory blanks were analyzed, and all the results were corrected according to the blank and recovery results.", was reworded as "A total of 8 field blanks and 26 laboratory blanks were analyzed, with individual blank values of N.D. (not dected)–1.1 pg/m3 and N.D.–1.3 pg/m3, respectively. All the results were corrected according to the blank and recovery results. " in lines 149–150 in the revised manuscript.

Query 8. Did the authors use the matrix spike? Is there any matrix effect in passive air samples? Response: To control and assurance the PFAAs analysis quality, except for strictly pre–cleaning of XAD and HPLC–MS/MS experimental operation, we also conducted internal standards recovery experiment, field blank experiment, and laboratory blank experiment. Results showed that the mean spiked PFAAs recoveries ranged from 81%±25% to 108%±22%, the field blanks and laboratory blanks values were N.D.–1.1 and N.D.–1.3 pg/m3, respectively, and all the results were corrected according to the blank and recovery results. Considering all these above results and several reported researches, the matrix spike experiment was not used in this research.

Special thanks to you for your good comments!

Reference Cardenas, A., Gold, D. R., Hauser, R., Kleinman, K. P., Hivert, M. F., Calafat, A. M., Ye, X., Webster, T. F., Horton, E. S., and Oken, E.: Plasma Concentrations of Per- and Polyfluoroalkyl Substances at Baseline and Associations with Glycemic Indicators and Diabetes Incidence among High-Risk Adults in the Diabetes Prevention Program Trial, Environ Health Perspect, 125, 107001, 2017. Hu, X. C., Andrews, D. Q., and Lindstrom, A. B.: Detection of Poly- and Perfluoroalkyl Substances (PFASs)in U.S. Drinking Water Linked to Industrial Sites, Military Fire TrainingAreas, and Wastewater Treatment Plants, Environ Sci Technol Lett, 3, 344-350, 2016. Karásková, P., Codling, G., Melymuk, L., and Klánová, J.: A critical assessment of passive air samplers for per- and polyfluoroalkyl substances, Atmos Environ, 185, 186-195, 2018. Sedlak, M. D., Benskin, J. P., Wong, A., Grace, R., and Greig, D. J.: Per- and polyfluoroalkyl substances (PFASs) in San Francisco Bay wildlife: Temporal trends, exposure pathways, and notable presence of precursor compounds, Chemosphere, 185, 1217-1226, 2017. Tian, Y., Zhou, Y., Miao, M., Wang, Z., Yuan, W., Liu, X., Wang, X., Wang, Z., Wen, S., and Liang, H.: Determinants of plasma concentrations of perfluoroalkyl and polyfluoroalkyl substances in pregnant women from a birth cohort in Shanghai, China, Environment International, 119, 165-173, 2018. Wang, Q. W., Yang, G. P., Zhang, Z. M., and Jian, S.: Perfluoroalkyl acids in surface sediments of the East China Sea, Environ Pollut, 231, 59-67, 2017. Dreyer, A., Kirchgeorg, T., Weinberg, I., Matthias, V. Particle-size distribution of airborne poly- and perfluorinated alkyl substances. Chemosphere ã129, 142-149, 2015.

We tried our best to improve the manuscript and made some changes in the manuscript. These changes will not influence the content and framework of the paper. We appreciate for Editors/ Reviewers' warm work earnestly, and hope that the correction will meet with approval. Once again, thanks very much for your comments and suggestions.

Yours sincerely,

Best regards!

Deming Han Doctoral Tel: +86 21 54743936 Fax: (86 21) 5474 0825 E-mail: han-deem@sjtu.edu.cn Add.:800 Dongchuan Road, Minhang District Shanghai, China

Please also note the supplement to this comment:
https://www.atmos-chem-phys-discuss.net/acp-2019-676/acp-2019-676-SC2-supplement.pdf

**Supplement:**

[revised manuscript text omitted]

---

## Author Comment (AC1) · 5 Sep 2019

A list of responses for comments from editors and reviewers

Dear Editors and Reviewers:

Thank you for your letter and for the reviewers' comments concerning our manuscript entitled "Occurrence and source apportionment of perfluoroalkyl acids (PFAAs) in the atmosphere in China" (Ref: acp–2019–676). These comments are valuable and very helpful for revising and improving our paper, as well as the important guiding significance to our researches. We have studied comments carefully and have made correction, the correction in the manuscript was marked–up with blue colour and underline (e.g. Revised Manuscript) which we hope meet with approval. The main corrections in

the paper and the responds to the reviewer's comments are as flowing: Responds to the editors' and reviewers' comments:
Major comment: Query 1. It is very unlikely that PFSAs occur in the gas phase. Recent studies have shown that PFSAs have been measured in passive samplers since passive samplers are also collecting particles. It is really important to clarify in this manuscript that PFSAs are typically particle–bound and the measured concentrations of PFSAs are most likely due to the collection of particles using the passive samplers. Make also clear that the measured concentration is NOT the gas phase but rather "air concentrations" with mainly gas phase and partly particulate phase. This has to be clarified at all places in the manuscript, figures and SI before a publication can be considered. Response: Thanks for the reviewer's hard work on reviewing our manuscript. We respect the reviewer's opinion and made the corresponding revision in the revised manuscript, however we didn't agree that PFSAs can not occur in the gas phase. PFSAs maybe could occur in the gas phase. Due to the lower acid dissociation coefficient (pKA), 0–3.8 for PFCAs and –3.3 for PFSAs, PFAAs are expected to be mainly associated with aerosols in the non–volatile anionic form (Lai et al., 2018;Pavlína et al., 2018). Additionally, in one research of particle–size distribution of airborne PFASs, PFOA was predominantly (>70%) observed in small size fraction (<0.14 $\mu$m), PFOS mass fractions were preferred to exist in the coarser size fractions (1.38–3.81 $\mu$m) (Dreyer et al., 2015). However, the occurrence of ionic PFAAs is not clear, more recent field studies have confirmed their occurrence in gaseous phase. For example, Fang et al., (2018) found that C2, C4–C10 PFCAs and C6 and C8 PFSAs were detected in the gas phase in the air above the Bohai and Yellow Seas, China, with total gaseous concentrations of 0.076–4.0 (0.77±0.97) pg/m3. Karásková et al., (2018) conducted an investigation via a active air sampler with quartz fiber filter and XAD impregnated sorbent based

[Figure]

PAS to capture particulate and gaseous PFAAs, found PFASs were primarily in the gas phase, with gaseous associated fractions of PFASs of 93%±96%; while PFCAs distribute between both particles and gas phase, with gaseous associated fraction values of 6%–98%. However, just as proposed by the reviewer that the XAD–PAS may collect particle bound PFAAs in this study. Despite several research suggest the collecting efficiency of particle PFAAs sample to be similar to gaseous samples, it is still difficult to distinguish them. According to reviewer's suggestion, it has clarified in this revised manuscript that PFSAs concentration are a combine of gaseous and particulate phases, at all places. Additionally, the description of "We should keep in mind that the unimpeded movement of particle bound PFAAs would be captured during sampling using XAD–PAS, which cannot differentiate PFAAs between gas and particle phases. Despite some research suggest the sampling efficiency of gas and particle phase PFAAs were similar (Karásková et al., 2018). In the present study, the two phases PFAAs sampled by XAD–PAS were treated as the whole atmosphere PFAAs concentration." in lines 100–104 in the revised manuscript.

General comments: Query 2. Use two significant numbers; Response: The number format has revised according to reviewer's suggestion.

Query 3. Table 1: It should be "HV–AAS" for Toronto, Canada; Response: Both HV–AAS and XAD impregnated sorbent based SIP–PAS samplers were used in the cited reference for Toronto, Canada. The PFASs concentrations sampled for these two samplers were different for their different sampling volume. The "XAD–PAS 0.7–20 pg/m3" has revised to "SIP–PAS 11.24±7.95 pg/m3" in Table 1 in the revised manuscript.

Query 4. Figure 3: Change to "Northwest"; Response: This mistake has been revised in Figure 3 in the new version of manuscript.

Figure R1. The revised Fig. 3

Query 5. Figure 4: Describe the four factors in the figure caption; Response: According to reviewer's suggestion, all the factor names were added in Figure 4 caption in the revised manuscript.

Figure R2. The revised Fig. 4 (The right figure is the revised figure with the corresponding figure caption)

Specific comments: Query 6. Line 13: Add total number of samples, number of sampling locations and which sampling method was used; Response: Thanks for reviewer's good suggestion. It has been revised in the new version manuscript in lines 17–20 in the revised manuscript, detailed description as "A nationwide geographical investigation considering atmospheric PFAAs via XAD–Passive Air Sampler was conducted in 23 different provinces/municipalities/autonomous regions in China, which provides an excellent chance to investigate their occurences, spatial trends, and potential sources. The total atmospheric concentrations of thirteen PFAAs (n=268) were 6.19–292.6 pg/m3,"

Query 7. Line 13: indicate if PFAAs were measured in the gas or particulate phase; Response: Just as discussed above, the PFAAs samples gathered via XAD–PAS should be a combine of gaseous and particulate phases. Hence, it was revised as "The total atmospheric concentrations of thirteen PFAAs (n=268) were 6.19–292.6 pg/m3," in lines 13–14 in the revised manuscript.

Query 8. Line 18: Specify which location are these "areas"; Response: According to reviewer's suggestion, "Spatially, the content of PFAAs displayed a declining gradient trend of central ares> eastern areas> western areas," has been changed to "Spatially, the content of PFAAs displayed a declining gradient trend of central of China> northern of China> eastern of China> northeast of China> southwest of China> northwest of China> southern of China areas, " in lines 18–20 in the revised manuscript.

Query 9. Line 24: Change to "ionizable"; Response: This mistake has been revised in line 26 in the revised manuscript.

Query 10. Lines 187–189: Clarify that this is the average of x numbers of sampling locations in China; Response: According to reviewer's suggestion, it was changed to "In general, an increasing seasonal mean of PFAAs concentrations from 23 sampling sites existed for summer (31.4 pg/m3) < autumn (35.6 pg/m3) < spring (42.4 pg/m3) < winter (52.8 pg/m3). " in lines 199–201 in the revised manuscript.

Query 11. Lines 216–218: Indicate, how many sites were included from each area "(n = : : :); Response: According to reviewer's suggestion, the description of "Overall, the predominant declining gradient of PFAAs' contents was CC> NC> EC> NE> SW> NW> SC areas in China,", was changed to "Overall, the predominant declining gradient of PFAAs' contents was CC (3 sites)> NC (3 sites)> EC (7 sites)> NE (2 sites)> SW (3 sites)> NW (3 sites)> SC (2 sites) areas in China, " in lines 228–229 in the revised manuscript.

Special thanks to you for your careful reading and good comments!

Reference Dreyer, A., Kirchgeorg, T., Weinberg, I., Matthias, V. Particle-size distribution of airborne poly- and perfluorinated alkyl substances. ChemosphereÂă129, 142-149, 2015. Fang, X., Wang, Q., Zhao, Z., Tang, J., Tian, C., Yao, Y., Yu, J., and Sun, H.: Distribution and dry deposition of alternative and legacy perfluoroalkyl and polyfluoroalkyl substances in the air above the Bohai and Yellow Seas, China, Atmos Environ, Karásková, P., Codling, G., Melymuk, L., and Klánová, J.: A critical assessment of passive air samplers for per- and polyfluoroalkyl substances, Atmos Environ, 185, 186-195, 2018. Lai, F. Y., Rauert, C., Gobelius, L., and Ahrens, L.: A critical review on passive sampling in air and water for per- and polyfluoroalkyl substances (PFASs), TrAC Trends in Analytical Chemistry, Available online 23 Nov. 2018. Pavlína, K., Garry, C., Lisa, M., and Jana, K.: A critical assessment of passive air samplers for per- and polyfluoroalkyl substances, Atmos Environ, 185, 186-195, 2018.

We tried our best to improve the manuscript and made some changes in the manuscript. These changes will not influence the content and framework of the paper. We appreciate for Editors/ Reviewers' warm work earnestly, and hope that the correction will meet with approval. Once again, thanks very much for your comments and suggestions.

Yours sincerely,

Best regards!

Deming Han Doctoral Tel: +86 21 54743936 Fax: (86 21) 5474 0825 E-mail: han-deem@sjtu.edu.cn Add.:800 Dongchuan Road, Minhang District Shanghai, China

Please also note the supplement to this comment:
https://www.atmos-chem-phys-discuss.net/acp-2019-676/acp-2019-676-AC1-supplement.pdf

[Figure]

Fig. 1.

[Figure]

**Fig. 2.**

**Supplement:**

[revised manuscript text omitted]

---

## Referee Comment (RC2) · Anonymous Referee #2 · 9 Sep 2019

General comments This study provides a nationwide dataset of PFAAs in the Chinese atmosphere. It included 23 sampling locations at which XAD-PAS were deployed for one year and samples were taken approximately every month. The results were evaluated with regard to tempo-spatial variations and sources attribution was done using correlations, Hysplit backward trajectories and a PMF receptor models.

As most available studies on PFAAs in the atmosphere are derived from single or only a few sampling sites, this nationwide study is of interest to the international community to better understand the atmospheric distribution of PFAAs. Additionally, China is a country of specific interest as large parts of the PFAS production were shifted from countries in Western Europe, the US and Japan to China and other Asian countries.

A major query refers to the description and discussion of the used sampling technique.

[Figure]

In different parts of the manuscript, it is stated that XAD-PAS collects representative portions of both the particle and the gas phase (line 62, line 191). However, it is reported in other publications that XAD-PAS collects primarily the gas phase (Lai et al., 2018; Melymuk et al., 2014). This difference should be discussed somewhere in the manuscript. Moreover, the comparison of the reported concentrations with measurements in other regions in section 3.1 can be skewed because of different sampling techniques and sampling media. If a comparison like this is done, the differences between the sampling techniques and their possible effects on the results should be discussed in a paragraph.

The manuscript is well structured and the reader can easily follow the drain of thoughts. However, it still contains several typing and grammar errors. Some are addressed in the section "technical corrections", but this is not exhaustive. Further proofreading by a native speaker would improve the manuscript.

Specific comments - The number of significant digits should be consistent throughout the manuscript.

Introduction - Line 24: PFASs include per- and polyfluoroalkyl substances and not only polyfluoroalkyl substances as stated in this line. - Line 32: In the PFAS community, usually the definition of Buck et al. (2011) is used to differentiate between short- and long-chain homologues. According to this, long-chain PFCAs possess 8 or more carbon atoms (7 perfluorinated carbon atoms plus the carboxy group) - Line 35/36: In May this year, the Parties to the Stockholm Convention adopted the listing of PFOA to Annex A. It would be good to add this new development to the text. - Line 63: Your references "Pavlina K et al., 2018" and "Karaskova P et al., 2018", used later in the manuscript, is in fact the same publication. Please change it to "Karaskova P et al, 2018" in the whole manuscript, as Karaskova (not Pavlina) is the family name of the author.

Material and Methods - Lines 85-86: Please add the number of sampling sites for each

of the seven divisions. - Line 87: It would be helpful for the reader to understand from Figure S1 which sampling site belongs to which region (NC, SC etc.). This information could be given in the map itself or in the figure caption. - Line 121: Usually, "A" refers to the aqueous phase and "B" to the organic solvent, not the other way round. It would avoid misunderstandings if this was turned around. - Line 126: There should be a reference to Table S3, which includes the mass transitions. - Line 134: Do the results refer to the linear isomer, e.g. of PFOS, or to the sum of all isomers? - Line 138: Please add the information, which PFAAs could be detected in which type of blanks and with which standard deviations, either in the text or in Table S3.

Results and Discussion - How are results below MDL given in this table? Does "0" refer to values below MDL? Please include this information. - For some of the results, the median value is below the MDL given in Table S3 (e.g. for PFTeDA). How did you calculate these median values? - Table 1: It would be helpful to know, which "PFAAs" are included in the sum given in the fifth column. - Line 199: It would be interesting which type of manufacturers are included in figure S3 and which industries are not? - Line 201 to 209: Was this monthly variation stronger for specific sampling sites than for others? - Line 272 to 274: It would be helpful for the reader to get a short explanation (1-2 sentences) why the air mass origins shown in figure S5 were a driving factor for PFAA variation. - Line 300: The production of PFOA to use it as emulsifier in PTFE manufacturing is also an important direct source in China, isn't it? - Line 331: You state in the conclusion that the measured PFAAs were "several times to several magnitudes higher" than other urban atmosphere levels. This is not that obvious when reading 3.1 and looking at table 1. For example, the values reported for Brno are in a similar range as the results from this study, if I understand it correctly?

Technical corrections - Line 15/16 "perfluorohexanoic" and "perfluoroheptanoic" have to be without "-" - Line 21: It has to be "fluorotelomer-based" instead of "fluoro-telomere-based" - Line 65: "deployed" instead of "depolyed" - Lines 85-86. I think it has to be "north of China (NC)" or "northern China (NC)" instead of "northern of China (NC)".

This also applies to the other regions. - Line 160: "which conducted in the landfill atmosphere in Tianjin" does not connect to the rest of the sentence. - Line 167: "neutral PFASs in Chinese air" instead of "neural PFASs in China air" - Line 189: "may be could attribute" is ungrammatical. - Line 318: correlations "to" each other

References Buck, R.C., Franklin, J., Berger, U., Conder, J.M., Cousins, I.T., de Voogt, P., van Leeuwen, S.P., 2011. Perfluoroalkyl and polyfluoroalkyl substances in the environment: terminology, classification, and origins. Integr. Environ. Assess. Manag. 7(4), 513-541. doi:10.1002/ieam.258. Lai, F., Rauert, C., Gobelius, L., Ahrens, L. (2018) A critical review on passive sampling in air and water for per- and polyfluoroalkyl substances (PFASs). TrAC Trends in Analytical Chemistry. https://doi.org/10.1016/j.trac.2018.11.009 Loewen, M., Wania, F., Wang, F., Tomy, G., 2008. Altitudinal Transect of Atmospheric and Aqueous Fluorinated Organic Compounds in Western Canada. Environ. Sci. Technol. 42(7), 2374-2379. https://doi.org/10.1021/es702276c.

---

## Short Comment (SC3) · 19 Sep 2019

Dear Reviewers #2:

Thank you for your comments concerning our manuscript entitled "Occurrence and source apportionment of perfluoroalkyl acids (PFAAs) in the atmosphere in China" (Ref: acp–2019–676). These comments are valuable and very helpful for revising and improving our paper, as well as the important guiding significance to our researches. We have studied comments carefully and have made correction, the correction in the manuscript was marked up with blue colour and underline (e.g. Revised Manuscript) which we hope meet with approval. The main corrections in the paper and the responds to the your comments are as flowing:
General comments. This study provides a nationwide dataset of PFAAs in the Chinese atmosphere. It included 23 sampling locations at which XAD-PAS were deployed for one year and samples were taken approximately every month. The results were evaluated with regard to tempo-spatial variations and sources attribution was done using correlations, Hysplit backward trajectories and a PMF receptor models. As most available studies on PFAAs in the atmosphere are derived from single or only a few sampling sites, this nationwide study is of interest to the international community to better understand the atmospheric distribution of PFAAs. Additionally, China is a country of specific interest as large parts of the PFAS production were shifted from countries in Western Europe, the US and Japan to China and other Asian countries. Response: Thanks for your appraisal for ou manuscript. We appreciate your valuable comments for improving our manuscript.

Major comment: Query (1). A major query refers to the description and discussion of the used sampling technique. In different parts of the manuscript, it is stated that XAD-PAS collects representative portions of both the particle and the gas phase (line 62, line 191). However, it is reported in other publications that XAD-PAS collects primarily the gas phase (Lai et al., 2018; Melymuk et al., 2014). This difference should be discussed somewhere in the manuscript. Response: Thanks for the reviewer's good suggestion. As reported by previous researches (Melymuk et al., 2014; Lai et al., 2018), XAD could sample solely gas-phase pollutants, while PUF is able to accumulate both gas-phase and particle associated semi-volatile pollutants, although the particles sampled with a low accuracy and variable sampling rates. However, due to the XAD-PAS sampler design, the atmospheric aerosols bound PFAAs could moved into the sampler. Especially for the aerosol size distributions of particle bound PFASs varied with individual specie, e.g. the airborne PFASs, PFOA was predominantly (>70%) observed in small
size fraction (<0.14  $\mu$ m) (Dreyer et al., 2015). In fact, Okeme et al., (2016) employed XAD-Pocket PAS to sample gaseous and particulate SVOCs in indoor environment, with finding not consistent with previous result that XAD-PAS could sample gas phase pollutant singly. And suggested that the sample efficiency of XAD sorbent sampler for gaseous and particulate phases pollutants need further investigations. Considering reviewer's suggestion, this differences were discussed and added, detailed as following: (1). The description of "However, recent field studies have confirmed their occurrence in gaseous phase (Lai et al., 2018;Cassandra et al., 2018;Ahrens et al., 2013)." in lines 57-58 in the original manuscript, was changed to "However, recent field studies have confirmed their occurrence in gaseous phase (Cassandra et al., 2018; Ahrens et al., 2013), e.g. Fang et al., (2018) found the total concentrations of C2, C4-C10 PFCAs and C6 and C8 PFSAs in the gas phase were 0.076–4.0 pg/m3 in the air above the Bohai and Yellow Seas, China." in lines 61-64 in the revised manuscript. (2). The description of "The particle size of XAD-2 is  $\sim$ 20-60 mesh, with water content of 20%-45%, its specific surface area  $>430 \text{ m}^2/\text{g}$ , and the reference adsorption capacity  $>35 \text{ m}^2/\text{g}$ . We should keep in mind that the unimpeded movement of particle bound PFAAs would be captured during sampling using XAD-PAS, which cannot differentiate PFAAs between gas and particle phases. Despite some research suggest the sampling efficiency of gas and particle phase PFAAs were similar (Karásková et al., 2018). In the present study, the two phases PFAAs sampled by XAD-PAS were treated as the whole atmosphere PFAAs concentration." was added in lines 106-111 in the revised manuscript. (3). "Fortunately, a number of reports showed that the XAD (a styrene-divinylbenzene copolymer) impregnated sorbent based passive air sampler (SIP-PAS) and XAD based PAS (XAD-PAS), were proven to be an ideal alternative sampling tool for monitoring PFAAs in a wide region, which was suggested to collect a representative sample of both gas and particle phases (Lai et al., 2018; Pavlína et al., 2018)." in lines 60-63 in the original manuscript, was changed to "Fortunately, a number of reports showed that the XAD (a styrene-divinylbenzene copolymer) impregnated sorbent based passive air sampler (SIP-PAS) and XAD based PAS (XAD-PAS), were proven to be an

**ACPD**
ideal alternative sampling tool for monitoring PFAAs in a wide region. Despite several publications suggested XAD-PAS collects primarily gaseous PFAAs (Melymuk et al., 2014; Lai et al., 2018) in the ambient, current findings were not consistent. Due to the unimpeded movements of particles into the sampler, XAD–PAS was indicated to collect a representative sample of both gas and particle phases (Ahrens et al., 2013; Okeme et al., 2016; Karásková et al., 2018). Moreover, the dominant sorbent for fluorinated compounds was reported as XAD resin in the XAD impregnated SIP–PAS, instead of PUF themselves (Krogseth et al., 2013)." in lines 65-73 in the revised manuscript.

Query (2). Moreover, the comparison of the reported concentrations with measurements in other regions in section 3.1 can be skewed because of different sampling techniques and sampling media. If a comparison like this is done, the differences between the sampling techniques and their possible effects on the results should be discussed in a paragraph. Response: According to reviewer's suggestion, the limitation of direct comparison between PFAAs concentration and other measurements was discussed and added in the revised manuscript, as following: "Although there existed inherent differences of PFAAs levels between regions, the impacts from differences in sampling techniques and sorbents between XAD-PAS and SIP-PAS could not be neglected. As indicated by previous researches, XAD has much higher sorptive capacity of PFASs than PUF, wind speed and temperature displayed different degrees of impact on their sampling capacity among different regions. Additionally, UV radiation has the potential to degradate PFAAs due to O3, OHÂů, and other atmospheric oxidants during sampling. " in lines 202-206 in the revised manuscript.

Query (3). The manuscript is well structured and the reader can easily follow the drain of thoughts. However, it still contains several typing and grammar errors. Some are addressed in the section "technical corrections", but this is not exhaustive. Further proof-reading by a native speaker would improve the manuscript. Response: Thanks for the reviewer's hard work on reviewing our manuscript. According to reviewer's suggestion, we have sent the revised manuscript to a professional English language editing service

**ACPD**
provider in science. This revised manuscript was revised carefully and checked line by line, numerous grammaticalÂămistakes and errors were corrected. For example, "to investigate their occurrences" in line 12 in the original manuscript, was reworded as "to investigate their occurrences" in line 13 in the revised manuscript; "was reported ranks as" in line 167 in the original manuscript, was changed to "was reported to rank as" in line 188 in the revised manuscript.

Query (4). Specific comments - The number of significant digits should be consistent throughout the manuscript. Response: According to reviewer's suggestion, the number of significant digits of concentrations were revised, and kept consistent throughout the manuscript.

Introduction Query (5). -Line 24: PFASs include per- and polyfluoroalkyl substances and not only polyfluoroalkyl substances as stated in this line. Response: As suggested by reviewer, the description of line 24 in the original manuscript was changed to "Perfluoroalkyl acids (PFAAs) are one class of ionic polyfluoroalkyl substances (PFASs), which have excellent characteristics in terms of chemical and thermal stability, high surface activity, and water and oil repulsion (Lindstrom et al., 2011;Wang et al., 2014)." in line 26 in the revised manuscript.

Query (6). -Line 32: In the PFAS community, usually the definition of Buck et al. (2011) is used to differentiate between short- and long-chain homologues. According to this, long-chain PFCAs possess 8 or more carbon atoms (7 perfluorinated carbon atoms plus the carboxy group). Response: According to reviewer's suggestion, the classification of long-chain and short-chain PFAAs homologues were revised based on study of Buck et al. (2011). The description of "Of the PFAAs, the long–chain (C  $\geq$ 7) perfluoroalkyl carboxylic acids (PFCAs) and (C  $\geq$ 6) perfluoroalkyl sulfonic acids (PFSAs) are more toxic and bio–accumulative than their short–chain analogues (Konstantinos et al., 2010)." in line 32 in the original manuscript, was reworded as "Of the PFAAs, the long–chain (C  $\geq$ 8) perfluoroalkyl carboxylic acids (PFCAs) and (C  $\geq$ 7) perfluoroalkyl sulfonic acids (PFSAs) are disting (PFSAs) are more toxic and bio–accumulative than their short–chain analogues (C  $\geq$ 8) perfluoroalkyl carboxylic acids (PFCAs) and (C  $\geq$ 7) perfluoroalkyl sulfonic acids (PFSAs) are disting (PFSAs) are more toxic and bio–accumulative than their short–chain analogues (C  $\geq$ 8) perfluoroalkyl carboxylic acids (PFCAs) and (C  $\geq$ 7) perfluoroalkyl sulfonic acids (PFSAs) are more toxic and bio–accumulative than their short–chain analogues (C  $\geq$ 8) perfluoroalkyl carboxylic acids (PFCAs) and (C  $\geq$ 7) perfluoroalkyl sulfonic acids (PFSAs) are more toxic and bio–accumulative than their short–chain analogues

ACPD
(Buck et al., 2011)." in lines 35-36 in the revised manuscript. Also, the corresponding result was revised, e.g. "To the contrary, a recent measurement found the long chain (C  $\geq$  8) PFCAs were much higher which conducted in the landfill atmosphere in Tianjin, China (Tian et al., 2018)." in lines 159-161 in the original manuscript, was changed to "Similarly, a recent PFAAs measurement conducted in the landfill atmosphere in Tianjin, China (Tian et al., 2018), found the long chain PFCAs were much higher than the short species." in lines 182-183 in the revised manuscript.

Query (7). -Line 35/36: In May this year, the Parties to the Stockholm Convention adopted the listing of PFOA to Annex A. It would be good to add this new development to the text. Response: According to reviewer's suggestion, the description of "This especially applies to perfluorooctanoic acid (PFOA) and perfluorohexane sulfonate (PFHxS) for which have been regulated in numerous countries, while perfluorooctane sulfonate (PFOS) have been added to Annex B of the Stockholm Convention in 2009 (Johansson et al., 2008)." in lines 34-36 in the original manuscript, was reworded as "This especially applies to perfluorooctanoic acid (PFOA), perfluorooctane sulfonate (PFOS) and perfluorohexane sulfonate (PFOS), in which PFOS and PFOA have been added to Annex A of the Stockholm Convention in 2009 and 2019, respectively, while PFHxS was under review by the Persistent Organic Pollutants Review Committee (Johansson et al., 2008; UNEP Stockholm Convention, 2019)." in lines 36-40 in the revised manuscript.

Query (8). -Line 63: Your references "Pavlina K et al., 2018" and "Karaskova P et al., 2018", used later in the manuscript, is in fact the same publication. Please change it to "Karaskova P et al, 2018" in the whole manuscript, as Karaskova (not Pavlina) is the family name of the author. Response: Thanks for the reviewer's hard work on reviewing our manuscript. The reference of "Pavlina K et al., 2018" was changed to "Karaskova P et al., 2018" in the revised manuscript, and the reference of "Pavlina K et al., 2018" was deleted in the revised manuscript.

Material and Methods Query (9). -Lines 85-86: Please add the number of sampling

**ACPD**
sites for each of the seven divisions. Response: Considering reviewer's suggestion, the description of "These sampling sites were divided into seven administrative divisions: norther of China (NC), southern of 86 China (SC), central of China (CC), eastern of China (EC), northwest of China (NW), northeast of China (NE), and southwest of China (SW)." in lines 85-87 in the original manuscript, was reworded as "These sampling sites were divided into seven administrative divisions: norther China (NC, n=3 sites), southern China (SC, n=2), central China (CC, n=3), eastern China (EC, n=7), northwest of China (NW, n=3), northeast of China (NE, n=2), and southwest of China (SW, n=3)." in lines 98-100 in the revised manuscript.

Query (10). -Line 87: It would be helpful for the reader to understand from Figure S1 which sampling site belongs to which region (NC, SC etc.). This information could be given in the map itself or in the figure caption. Response: Considering reviewer's suggestion, the information of each sampling site belonging to which region was added in Figure S1 the revised manuscript. Detailed revision was as following: Figure R1. Revised figure S1 in the manuscript, the upper for the original one, the bottom for the revised figure.

Query (11). -Line 121: Usually, "A" refers to the aqueous phase and "B" to the organic solvent, not the other way round. It would avoid misunderstandings if this was turned around. Response: Considering reviewer's suggestion, the description of "The gradient elution program of the mobile phase A (methanol) and B (5 mmol/L aqueous ammonium acetate) was 20% A + 80% B at the start, 95% A + 5% B at 8 min, 100% a at 13 min, 20% A + 80% B at 14 min, and was maintained for 6 min." in lines 120-122 in the original manuscript, was reworded as "The gradient elution program of the mobile phase A (5 mmol/L aqueous ammonium acetate) and B (methanol) was 80% A + 20% B at the start, 5% A + 95% B at 8 min, 100% a at 13 min, 80% A + 20% B at 14 min, and was maintained for 6 min." in lines 138-140 in the revised manuscript.

Query (12). -Line 126: There should be a reference to Table S3, which includes the mass transitions. Response: Considering reviewer's suggestion, references of "

**ACPD**
Karásková et al., 2018" and "Liu et al., 2015" were added to this table in the revised manuscript.

Query (13). -Line 134: Do the results refer to the linear isomer, e.g. of PFOS, or to the sum of all isomers? Response: Thanks for reviewer's good suggestion, this result refer to the liner isomer.

Query (14). -Line 138: Please add the information, which PFAAs could be detected in which type of blanks and with which standard deviations, either in the text or in Table S3. Response: Considering reviewer's suggestion, the detailed information of filed blanks and laboratory blanks was added in the revised Table S3 in the revised manuscript. Table R1 (Table S3). MS parameters, MDLs, LODs, LOQs values, recovery rates and blank values for individual compounds of PFAAs

Results and Discussion Query (15). - How are results below MDL given in this table? Does "0" refer to values below MDL? Please include this information. Response: Thank very much for reviewer's suggestion. The measured abundances of PFAAs which below MDL was marked as "0" in the original manuscript. Considering reviewer's suggestion and some statistics standard used, we have modified these values to "BDL" in Table S4 in the revised Supporting Materials, as following: Table R2 (Table S4). The measured abundances of PFAAs in this studyïijĹn=268ïijL'

Query (16). - For some of the results, the median value is below the MDL given in Table S3 (e.g. for PFTeDA). How did you calculate these median values? Response: Thanks for the reviewer's suggestion. WeÂăvalueÂăthisÂăsuggestionÂăvery highly, checked these results carefully and found the MDL of PFTeDA was 0.14 rather 0.41, and revised this mistake. For the statistics analysis of measured concentrations, the results of BDL were replaced by 1/2 of the corresponding MDL values. Considering reviewer's suggestion, the description of "Statistical analyses were carried out by SPSS Statistics 22 (IBM Inc. US) and SigmaPlot 14.0 (Systat Software, US)." in line 147 in the original manuscript, was changed to "Statistical analyses were carried out by SPSS
Statistics 22 (IBM Inc. US), and the values of 1/2 MDL were used to replace these measured results of BDL. The statistics figures were depicted using technical software of SigmaPlot 14.0 (Systat Software, US)." in lines 166-168 in this revised manuscript.

Query (17). -Table 1: It would be helpful to know, which "PFAAs" are included in the sum given in the fifth column. Response: Considering reviewer's suggestion, more detailed information on species PFAAs was added in the note of "b" of PFAAs in Table 1, it was reworded as "b: represent the total concentration ranges of PFCAs and PF-SAs; mean concentrations of the total PFCAs and PFSAs;" in Table 1 in the revised manuscript.

Query (18). -Line 199: It would be interesting which type of manufacturers are included in figure S3 and which industries are not? Response: Considering reviewer's suggestion, more detailed information on the fluoride manufacturer was added to the caption of Figure S3 in the revised Supporting Materials, "Figure S3. The spatial distributions of fluoride related products manufacturers in China (note that part of fluoride related industries were not included in this figure) and the different geographical conditions" "Figure S3. The spatial distributions of fluoride related products manufacturers in China and the different geographical conditions (note that the fluoride related manufacturers including textiles, crude plastic, paint coating, packaging materials, while part of fluoride related industries were not included in this figure)"

Query (19). -Line 201 to 209: Was this monthly variation stronger for specific sampling sites than for others? Response: The monthly variations of PFAAs varied based on site environments and local geographical conditions, the monthly variations of PFAAs in each site differed more or less. As shown in Figure S2, Beijing, Tianjin, and Xinjiang sites shared a similar monthly PFAAs variations, while Shanghai displayed a much different trend, which was controlled by local sources emissions as well.

Query (20). -Line 272 to 274: It would be helpful for the reader to get a short explanation (1-2 sentences) why the air mass origins shown in figure S5 were a driving
factor for PFAA variation. Response: Considering reviewer's suggestion, an explanation of the air mass origins in Table S5 was added in the manuscript, as "As illustrated in Figure S5, the 48 hours back trajectories were generally associated with air masses originating from the surrounding areas of the sampling locations, the trajectories which overlapped with urban areas in Zhejiang, Jiangxi and Shanghai, which confirmed that the air mass origins was a driving factor for PFAAs variation. " in lines 298-301 in the revised manuscript.

Query (21). -Line 300: The production of PFOA to use it as emulsifier in PTFE manufacturing is also an important direct source in China, isn't it? Response: We strongly agree with reviewer's suggestion, since PFOA is widely used in the manufacturing of polytetrafluoroethylene (PTFE), perfluorinated ethylene propolymer (FEP), and perfluoroalkoxy polymers (PFA). Considering reviewer's suggestion, the description of "PFOA was considered as the marker for the emulsification of plastics, rubber products, flame retardants for textiles, paper surface treatments, and fire foams (Liu et al., 2015;Konstantinos et al., 2010)." in lines 299-300 in the original manuscript, was reworded as "PFOA was considered as the marker for the emulsification of plastics, rubber products, flame retardants for textiles, paper surface treatments, fire foams and PTFE emulsifiers (Liu et al., 2015;Konstantinos et al., 2010)." in lines 326-237 in the revised manuscript.

Query (22). -Line 331: You state in the conclusion that the measured PFAAs were "several times to several magnitudes higher" than other urban atmosphere levels. This is not that obvious when reading 3.1 and looking at table 1. For example, the values reported for Brno are in a similar range as the results from this study, if I understand it correctly? Response: Considering reviewer's suggestion, this description of this conclusion was modified, "Results indicated that the measured PFAAs were several times to several magnitudes higher than other urban atmosphere levels, and much higher abundances existed in winter seasons compared with in the summer." in lines 330-332 in the original manuscript, was changed to "Results indicated that the measured PFAAs in the present study were several times to several magnitudes higher than other urban atmosphere levels, and much higher abundances existed in winter seasons compared with in the summer." in lines 330-332 in the original manuscript, was changed to "Results indicated that the measured PFAAs in the present study were several times to several magnitudes higher than the levels indicated that the measured PFAAs in the present study were several times to several magnitudes higher than the levels
conducted in most other urban locations, while far lower than the measurements implemented at point sources. In which, the C5–C14 PFCAs analogues occupied 79.6% of the total PFAAs variations, PFOA, PFHxA and PFOS ranked the top three species. Additionally," in lines 357-361 in the revised manuscript.

Technical corrections Query (23). - Line 15/16 "perfluorohexanoic" and "perfluoroheptanoic" have to be without "-" Response: According to reviewer's suggestion, the "perfluoro–hexanoic acid (PFHxA)", and "perfluoro–heptanoic acid (PFPeA)" was revised as "perfluoro–hexanoic acid (PFHxA)" and "perfluoro–heptanoic acid (PFPeA)" in the revised manuscript, respectively.

Query (24). -Line 21: It has to be "fluorotelomer-based" instead of "fluoro-telomere based" Response: Considering reviewer's suggestion, "fluoro-telomere based" in line 21 in the original manuscript was reworded as "fluorotelomer-based" in line 22 in the revised manuscript.

Query (25). - Line 65: "deployed" instead of "depolyed" Response: Considering reviewer's suggestion, "depolyed" in line 65 in the original manuscript was reworded as "deployed" in line 74 in the revised manuscript.

Query (26). -Lines 85-86. I think it has to be "north of China (NC)" or "northern China (NC)" instead of "northern of China (NC)". This also applies to the other regions. Response: Considering reviewer's suggestion, these related description were modified. For example, in the section of "2.2 Sample Collection", "These sampling sites were divided into seven administrative divisions: norther of China (NC), southern of China (SC), central of China (CC), eastern of China (EC), northwest of China (NW), northeast of China (NE), and southwest of China (SW)." in line 85-87 in the original manuscript, was changed to "These sampling sites were divided into seven administrative divisions: norther China (NC, n=3 sites), southern China (SC, n=2), central China (CC, n=3), eastern China (EC, n=7), northwest of China (NW, n=3), northeast of China (NE, n=2), and southwest of China (SW, n=3)." in lines 98-100 in the revised manuscript. In ab-

**ACPD**
stract, the description of "Spatially, the content of PFAAs displayed a declining gradient trend of central areas > eastern areas > western areas, and Henan contributed as the largest proportion of PFAAs." in lines 17-19 in the original manuscript, was changed to "Spatially, the content of PFAAs displayed a declining gradient trend of central China> northern China> eastern China> northeast of China> southwest of China> northwest of China> southern China areas, and Henan contributed as the largest proportion of PFAAs." in lines 18-20 in the revised manuscript.

Query (27). -Line 160: "which conducted in the landfill atmosphere in Tianjin" does not connect to the rest of the sentence. Response: Considering reviewer's suggestion, "To the contrary, a recent measurement found the long chain (C  $\geq$  8) PFCAs were much higher which conducted in the landfill atmosphere in Tianjin, China (Tian et al., 2018)." in lines 159-161 in the original manuscript, was changed to "Similarly, a recent PFAAs measurement conducted in the landfill atmosphere in Tianjin, China (Tian et al., 2018), found the long chain PFCAs were much higher than the short species." in lines 182-183 in the revised manuscript.

Query (28). -Line 167: "neutral PFASs in Chinese air" instead of "neural PFASs in China air" Response: Considering reviewer's suggestion, "Meanwhile, one major variation of PFOA precursor, 8:2 FTOH, was reported ranks as the highest concentration among neural PFASs in China air" in lines 166-167 in the original manuscript, was changed to "Meanwhile, one major variation of PFOA precursor, 8:2 FTOH, was reported to rank as the highest concentration among neural PFASs in air of China" in lines 189-190 in the revised manuscript.

Query (29). -Line 189: "may be could attribute" is ungrammatical. Response: Considering reviewer's suggestion, "The winter maxima abundance of PFAAs may be could attribute to the stagnant atmospheric conditions," in lines 189-190 in the original manuscript, was changed to "The winter maxima abundance of PFAAs could be attribute to the stagnant atmospheric conditions," in lines 215-216 in the revised manuscript.
Query (30). -Line 318: correlations "to" each other Response: Considering reviewer's suggestion, the description of "In addition, these four analogues showed apparent positive correlations each other (r =0.59–0.79, p

C., Lisa, M., and Jana, K.: A critical assessment of passive air samplers for per- and polyfluoroalkyl substances, Atmos Environ, 185, 186-195, 2018.

We tried our best to improve the manuscript and made some changes in the manuscript. These changes will not influence the content and framework of the paper. We appreciate for Editors/ Reviewers' warm work earnestly, and hope that the correction will meet with approval. Once again, thanks very much for your comments and suggestions.

Yours sincerely,

Best regards!

Deming Han Ph.D Tel: +86 21 54743936 Fax: (86 21) 5474 0825 E-mail: handeem@sjtu.edu.cn Add.:800 Dongchuan Road, Minhang District Shanghai, China

Please also note the supplement to this comment: https://www.atmos-chem-phys-discuss.net/acp-2019-676/acp-2019-676-SC3supplement.pdf

**ACPD**
Figure R1. Revised figure S1 in the manuscript, the upper for the original one, the bottom for the revised figure.

**Supplement:**

[revised manuscript text omitted]

---

## Author Response (AR1)

**Anonymous Referee #RC1**

**Major comment:**

**Query 1. It is very unlikely that PFSAs occur in the gas phase. Recent studies have shown that PFSAs have been measured in passive samplers since passive samplers are also collecting particles. It is really important to clarify in this manuscript that PFSAs are typically particle–bound and the measured concentrations of PFSAs are most likely due to the collection of particles using the passive samplers. Make also clear that the measured concentration is NOT the gas phase but rather "air concentrations" with mainly gas phase and partly particulate phase. This has to be clarified at all places in the manuscript, figures and SI before a publication can be considered.**

*Response:* We want to thank reviewer #RC1 for the valuable feedback.

We respect the reviewer's opinion, while we didn't entirely agree with that PFSAs can not occur in the gas phase. Despite that due to the lower acid dissociation coefficient ($pK_A$), 0–3.8 for PFCAs and –3.3 for PFSAs, PFAAs are expected to be mainly associated with aerosols in the non–volatile anionic form (Lai et al., 2018;Pavlína et al., 2018). The airborne PFASs, PFOA was predominantly (>70%) observed in small size fraction (<0.14 μm), PFOS mass fractions were preferred to exist in the coarser size fractions (1.38–3.81 μm) (Dreyer et al., 2015). However, the exact occurrence of ionic PFAAs is not clear till now, more recent field studies have confirmed their occurrence in gaseous phase. For example, Fang et al., (2018) found that C2, C4–C10 PFCAs and C6 and C8 PFSAs were detected in the gas phase in the air above the Bohai and Yellow Seas, China, with total gaseous concentrations of 0.076–4.0 (0.77±0.97) pg/m3. Karásková et al., (2018) conducted an investigation via a active air sampler with quartz fiber filter and XAD impregnated sorbent based PAS to capture particulate and gaseous PFAAs, found PFASs were primarily in the gas phase, with gaseous associated fractions of PFASs of 93%±96%; while PFCAs distribute between both particles and gas phase, with gaseous associated fraction values of 6%–98%.

It also should be noticed that just as proposed by the reviewer, the XAD–PAS could collect particle bound PFAAs in the present study. Though several research suggest the collecting efficiency of particle PFAAs sample to be similar to gaseous samples, it is still difficult to distinguish them. **Hence, we agree with the reviewer that the**

**measured PFAAs concentrations should be the combination of gaseous and particulate PFAAs concentrations rather than the gas phase alone. According to reviewer's suggestion, we improved the description of atmospheric PFSAs concentrations in the revised manuscript.** The atmospheric PFSAs concentrations were clarified as a combine of gaseous and particulate phases, and were revised at all places in the revised manuscript, figures and supporting materials.

Moreover, to achieve a better understanding of this for the potential readers, the original description of "However, recent field studies have confirmed their occurrence in gaseous phase (Cassandra et al., 2018;Ahrens et al., 2013).", was reworded as "However, recent field studies have confirmed their occurrence in gaseous phase (Cassandra et al., 2018;Ahrens et al., 2013), e.g. Fang et al., (2018) found the total concentrations of C2, C4–C10 PFCAs and C6 and C8 PFSAs in the gas phase were 0.076–4.0 pg/m$^3$ in the air above the Bohai and Yellow Seas, China." in lines 61-64 in the revised manuscript.

We added description of "We should keep in mind that the unimpeded movement of particle bound PFAAs would be captured during sampling using XAD–PAS, which cannot differentiate PFAAs between gas and particle phases. Despite some research suggest the sampling efficiency of gas and particle phase PFAAs were similar (Karásková et al., 2018). In the present study, the reported PFAAs sampled by XAD-PAS represent a combination of gaseous and particulate PFAAs concentration." in lines 107–111 in the revised manuscript.

**General comments:**

**Query 2.** Use two significant numbers;

*Response:* The number format of reported PFAAs concentrations have been all revised according to reviewer's suggestion in the revised manuscript, all the concentration values modified to two significant numbers and kept consistent throughout the manuscript.

**Query 3.** Table 1: It should be "HV–AAS" for Toronto, Canada;

*Response:* Considering reviewer's suggestion, we checked this reference again. Both HV–AAS and XAD impregnated sorbent based SIP–PAS samplers were used in the cited reference of Toronto, Canada, but the PFASs concentrations sampled for these two samplers were different for their different sampling volume. The "XAD–PAS 0.7–20 pg/m$^3$" has revised to "SIP–PAS 11.24±7.95 pg/m$^3$" in Table 1 in the revised manuscript.

**Query 4.** Figure 3: Change to "Northwest";

*Response:* We thanks very much for reviewer's careful reading our manuscript. This mistake has been revised in Figure 3 in the new version of manuscript, as following:

[Figure]

Figure R1. The revised Fig. 3 (The upper figure is the original one, while the bottom figure was the revised one)

**Query 5.** Figure 4: Describe the four factors in the figure caption;

*Response:* According to reviewer's suggestion, all the factor names were added in Figure 4 caption in the revised manuscript.

[Figure]

Figure R2. The revised Fig. 4 (The right figure is the revised figure with the corresponding figure caption)

**Specific comments:**

**Query 6.** Line 13: Add total number of samples, number of sampling locations and which sampling method was used;

*Response:* Thanks for reviewer's good suggestion. The original description of "A nationwide geographical investigation considering atmospheric PFAAs was conducted in China, which provides an excellent chance to investigate their occurences, spatial trends, and potential sources. The total concentrations of thirteen PFAAs were 6.19 – 292.6 pg/m³, with an average value of 39.8 ± 28.1 pg/m³, which were higher than other urban levels but lower than point source measurements.", has been revised in the new version manuscript in lines 11-15 in the revised manuscript, detailed description as "A nationwide geographical investigation considering atmospheric PFAAs via XAD–Passive Air Sampler was conducted in 23 different provinces/municipalities/autonomous regions in China, which provides an excellent chance to investigate their occurrences, spatial trends, and potential sources. A nationwide geographical investigation considering atmospheric PFAAs via XAD–Passive Air Sampler was conducted in 23 different provinces/municipalities/autonomous regions in China, which provides an excellent chance to investigate their occurences, spatial trends, and potential sources. The total atmospheric concentrations of thirteen PFAAs (n=268) were 6.19–292.6 pg/m³,"

**Query 7.** Line 13: indicate if PFAAs were measured in the gas or particulate phase;

*Response:* Just as discussed above, the PFAAs samples gathered via XAD–PAS should be a combine concentrations of gaseous and particulate phases PFAAs. Hence, it was revised as "The total atmospheric concentrations of thirteen PFAAs (n=268) were 6.19–292.6 pg/m$^3$," in lines 13–14 in the revised manuscript.

**Query 8.** Line 18: Specify which location are these "areas";

*Response:* According to reviewer's suggestion, "Spatially, the content of PFAAs displayed a declining gradient trend of central ares> eastern areas> western areas," has been changed to "Spatially, the content of PFAAs displayed a declining gradient trend of central of China> northern China> eastern China> northeast of China> southwest of China> northwest of China> southern China areas, " in lines 18–20 in the revised manuscript.

**Query 9.** Line 24: Change to "ionizable";

*Response:* After careful discussion, we still think "ionic perfluoroalkyl acids" was more suitable than "ionizable perfluoroalkyl acids", since PFAAs were regarded as divided into two subgroups ionic PFAAs and neutral PFAAs in most published papers.

**Query 10.** Lines 187–189: Clarify that this is the average of x numbers of sampling locations in China;

*Response:* According to reviewer's suggestion, the original description of "In general, an increasing 188 seasonal mean of PFAAs concentrations existed for summer (31.4 pg/m$^3$) < autumn (35.6 pg/m$^3$) < spring (42.4 pg/m$^3$) <189 winter (52.8 pg/m$^3$).", was changed to "In general, an increasing seasonal mean of PFAAs concentrations from 23 sampling sites existed for summer (31.4 pg/m$^3$) < autumn (35.6 pg/m$^3$) < spring (42.4 pg/m$^3$) < winter (52.8 pg/m$^3$). " in lines 214–216 in the revised manuscript.

**Query 11.** Lines 216–218: Indicate, how many sites were included from each area "(n = ∶ ∶ ∶);

*Response:* According to reviewer's suggestion, the description of sampling sites in each area was modified in front of the revised manuscript, modified as "These sampling sites were divided into seven administrative divisions: norther China (NC, n=3 sites), southern China (SC, n=2), central China (CC, n=3), eastern China (EC, n=7), northwest of China (NW, n=3), northeast of China (NE, n=2), and southwest of China (SW, n=3)." in lines 98–100 in the revised manuscript.

**Special thanks to you for your careful reading and good comments!**
**A national scale passive air sampling campaign was carried out in China, and 11 perfluoroalkyl acids compounds were determined in air samples during a whole year. The authors discussed concentration profiles, distributions and potential sources. The manuscript has the potential to add to the available body of evidence. I believe that the data are reliable and useful. In general, I recommend that the manuscript be accepted pending some minor revisions as outlined below.**

*Response:* Thank you for reviewer's appraisal of our manuscript. We appreciate reviewer's valuable comments for improving the manuscript.

**Several minor revisions:**

**Query 1. There should be a space between numbers and units, like line 96 and line 130, –20 $^o$C.**

*Response:* Thanks for reviewer's suggestion. The format of number and units in lines 96 and 130 were revised in the revised manuscript, and the others were reworded throughout the revised manuscript.

**Query 2. Line 29, the authors listed the environment like atmosphere, water, or snow, or in wildlife and even in the human body, however, the references cited seemed not match.**

*Response:* Considering reviewer's suggestion, the references cited were revised. The description of "PFAAs can change adult thyroid hormone levels, reduce newborn birth weight, and biomagnify in the food chain, which can be extremely toxic to animals and humans (Hu et al., 2016;Jian et al., 2017;Baard Ingegerdsson et al., 2010).", was changed as "PFAAs can be released to the surrounding environment during manufacturing and use of PFAAs containing products, which are ubiquitous in the environment (e.g., in the atmosphere, water, or snow) (Dreyer et al., 2009;Hu et al., 2016;Wang et al., 2017), in wildlife (Sedlak et al., 2017), and even in the human body (Cardenas et al., 2017;Tian et al., 2018)." in lines 30–33 in the revised manuscript.

**Query 3.** Line 32, the long–chain perfluoroalkyl carboxylic acids should be defined as C ≥ 8.

*Response:* According to reviewer's suggestion, the classification of long-chain and short-chain PFAAs homologues were revised. The description of "Of the PFAAs, the long–chain (C ≥7) perfluoroalkyl carboxylic acids (PFCAs) and (C ≥6) perfluoroalkyl sulfonic acids (PFSAs) are more toxic and bio–accumulative than their short–chain analogues (Konstantinos et al., 2010)." in line 32 in the original manuscript, was reworded as "Of the PFAAs, the long–chain (C ≥8) perfluoroalkyl carboxylic acids (PFCAs) and (C ≥7) perfluoroalkyl sulfonic acids (PFSAs) are more toxic and bio–accumulative than their short–chain analogues (Buck et al., 2011)." in lines 35-36 in the revised manuscript.

**Query 4.** Line 74, the PFCAs analogues abbreviations listed in brackets should be given the full name, because some of them occurred at the first time.

*Response:* All of full names and abbreviations of the 13 PFAAs analogues, were listed in Table S1 in the Supporting Materials. Considering reviewer's suggestion, the description of "The PFAAs standards used were Wellington Laboratories (Guelph, ON, Canada) PFAC–MXB standard materials, including C5–C14 PFCAs analogues (PFPeA, PFHxA, PFHpA, PFOA, PFNA, PFDA, PFUdA, PFDoA, PFTrDA, and PFTeDA), as well as C4, C6, and C8 PFSAs analogues (PFBS, PFHxS, and PFOS)." was changed to "The PFAAs standards used were Wellington Laboratories (Guelph, ON, Canada) PFAC–MXB standard materials, including C5–C14 PFCAs analogues (Perfluoropentanoic acid (PFPeA), Perfluorohexanoic acid (PFHxA), Perfluoroheptanoic acid (PFHpA), PFOA, Perfluorononanoic acid (PFNA), Perfluorodecanoic acid (PFDA), Perfluoroundecanoic acid (PFUdA), Perfluorododecanoic acid (PFDoA), Perfluorotridecanoic acid (PFTrDA), and Perfluorotetradecanoic acid (PFTeDA)), as well as C4, C6, and C8 PFSAs analogues (Perfluorobutane sulfonic acid (PFBS), PFHxS, and PFOS)." in lines 83-88 in the revised manuscript.

**Query 5. Why did not the authors collect all the samples from urban area?**

*Response:* This research is a part of a large study aimed to investigate the occurrence and regional transportation of new emerging pollutants in China, in which PFAAs was one crucial pollutant. This investigation was implemented by Shanghai Jiao Tong University and Shanghai Academic of Environmental Science, and the sampling sites were selected based on the comprehensive effects of sampling geographical location, availability of volunteers, convenience of exchanging sorbent of XAD–PAS, and some other factors. Therefore, 20 urban sampling sites and 3 rural sampling sites were selected in this investigation ultimately.

**Query 6.** **Please give the information on Amberlite XAD–2 resin.**

*Response:* Considering reviewer's suggestion, detailed information on XAD–2 was added in lines 106-107 in the revised manuscript, as "The particle size of XAD–2 is ~20–60 mesh, with water content of 20%–45%, its specific surface area ≥430 $m^2$/g, and the reference adsorption capacity ≥35 mg/g."

**Query 7.** **If the MDL was derived from three times SD of the field blank values, the authors should give the information about the field blanks and laboratory blanks. Which compounds were detected in those blanks? And in what level?**

*Response:* According to reviewer's suggestion, more information about filed blanks and laboratory blanks was added in the revised manuscript and supporting materials. For instance, the description of "A total of 8 field blanks and 26 laboratory blanks were analyzed, and all the results were corrected according to the blank and recovery results.", was reworded as "A total of 8 field blanks and 26 laboratory blanks were analyzed, with individual blank values of BDL (below detection limit)–1.12 $pg/m^3$ and BDL–1.29 $pg/m^3$, respectively. All the results were corrected according to the blank and recovery results. " in lines 157-160 in the revised manuscript. More detailed information on filed ad laboratory blanks values were added in Table S3, as following Table R1:

**Table R1 (Table S3)**. MS parameters, MDLs, LODs, LOQs values, recovery rates and blank values for individual compounds of PFAAs

| Analogues | Parent ions (m/z) | Daughter ion (m/z) | Declustering potential (V) [a] | Collision energy (eV) [b] | Retention time (s) | MDLs (pg/m$^3$) | LODs (pg/m$^3$) | LOQs (pg/m$^3$) | Recovery rate (%) | Filed bank (pg/m$^3$) | Laboratory blank (pg/m$^3$) | Internal Standards |
|---|---|---|---|---|---|---|---|---|---|---|---|---|
| PFCAs | | | | | | | | | | | | |
| PFPeA | 263 | 219 | -40 | –34 | 3.16 | 0.41 | 0.31 | 1.05 | 96±17 | 0.41±0.14 | 0.22±0.17 | 1,2–$^{13}$C$_2$–PFHxA |
| PFHxA | 313 | 269 | -35 | –36 | 3.42 | 0.18 | 0.14 | 0.47 | 108±22 | 0.48±0.06 | 0.37±0.39 | 1,2–$^{13}$C$_2$–PFHxA |
| PFHpA | 363 | 319→169 | -55 | –28 | 3.70 | 0.22 | 0.16 | 0.55 | 93±16 | 0.62±0.07 | 0.22±0.32 | 1,2,3,4–$^{13}$C$_4$–PFOA |
| PFOA | 413 | 369→169 | -45 | –39 | 3.99 | 0.33 | 0.26 | 0.87 | 91±13 | 0.93±0.11 | 0.41±0.29 | 1,2,3,4–$^{13}$C$_4$–PFOA |
| PFNA | 463 | 419→219 | -40 | –44 | 4.32 | 0.61 | 0.46 | 1.53 | 89±17 | 0.57±0.20 | 0.20±0.25 | 1,2,3,4,5–$^{13}$C$_5$–PFNA |
| PFDA | 513 | 469→219 | -50 | –47 | 4.67 | 0.56 | 0.42 | 1.39 | 93±11 | 0.35±0.19 | 0.28±0.22 | 1,2–$^{13}$C$_2$–PFDA |
| PFUdA | 563 | 519→269 | -45 | –61 | 5.02 | 0.28 | 0.21 | 0.70 | 88±16 | 0.31±0.09 | 0.31±0.13 | 1,2–$^{13}$C$_2$–PFUdA |
| PFDoA | 613 | 569→169 | -45 | –65 | 5.35 | 0.28 | 0.21 | 0.70 | 94±18 | 0.44±0.09 | 0.15±0.18 | 1,2–$^{13}$C$_2$–PFDoA |
| PFTrDA | 663 | 619→169 | -50 | –59 | 5.64 | 0.34 | 0.26 | 0.87 | 102±17 | 0.09±0.11 | 0.05±0.11 | 1,2–$^{13}$C$_2$–PFDoA |
| PFTeDA | 713 | 669→169 | -65 | –57 | 5.94 | 0.14 | 0.31 | 1.03 | 97±21 | 0.12±0.14 | 0.06±0.13 | 1,2–$^{13}$C$_2$–PFDoA |
| PFSAs | | | | | | | | | | | | |

| | | | | | | | | | | | |
|---|---|---|---|---|---|---|---|---|---|---|---|
| PFBS | 299 | 80→99 | -45 | –64 | 3.19 | 0.25 | 0.20 | 0.66 | 81±25 | 0.11±0.08 | 0.27±0.46 | $^{18}O_2$–PFHxS |
| PFHxS | 399 | 80→99 | -55 | –87 | 3.70 | 0.16 | 0.12 | 0.40 | 86±13 | 0.16±0.05 | 0.42±0.27 | $^{18}O_2$–PFHxS |
| PFOS | 499 | 80→99 | -55 | –98 | 4.31 | 0.24 | 0.19 | 0.63 | 95±15 | 0.75±0.08 | 0.54±0.61 | 1,2,3,4–$^{13}C_4$–PFOS |

Internal Standards

| | | | | | | | | | | | |
|---|---|---|---|---|---|---|---|---|---|---|---|
| 1,2–$^{13}C_2$–PFHxA | 315 | 270 | -75 | –41 | 3.40 | / | / | / | / | / | / | / |
| 1,2,3,4–$^{13}C_4$–PFOA | 417 | 372 | -40 | –41 | 3.99 | / | / | / | / | / | / | / |
| 1,2,3,4,5–$^{13}C_5$–PFNA | 468 | 423 | -84 | –52 | 4.34 | / | / | / | / | / | / | / |
| 1,2–$^{13}C_2$–PFDA | 515 | 470 | -87 | –51 | 4.69 | / | / | / | / | / | / | / |
| 1,2–$^{13}C_2$–PFUdA | 565 | 520 | -79 | –61 | 5.02 | / | / | / | / | / | / | / |
| 1,2–$^{13}C_2$–PFDoA | 615 | 570 | -66 | –55 | 5.35 | / | / | / | / | / | / | / |
| $^{18}O_2$–PFHxS | 403 | 103 | -55 | 97 | 3.72 | / | / | / | / | / | / | / |
| 1,2,3,4–$^{13}C_4$–PFOS | 503 | 80 | -80 | 97 | 4.31 | / | / | / | / | / | / | / |

**Query 8.** **Did the authors use the matrix spike? Is there any matrix effect in passive air samples?**

*Response:* To control and assurance the PFAAs analysis quality, except for strictly pre–cleaning of XAD and HPLC–MS/MS experimental operation, we also conducted internal standards recovery experiment, field blank experiment, and laboratory blank experiment. Results showed that the mean spiked PFAAs recoveries ranged from 81%±25% to 108%±22%, the field blanks and laboratory blanks values were N.D.–1.1 and N.D.–1.3 $pg/m^3$, respectively, and all the results were corrected according to the blank and recovery results. Considering all these above results and several reported researches, the matrix spike experiment was not used in this research.

**Special thanks to you for your good comments!**
**This study provides a nationwide dataset of PFAAs in the Chinese atmosphere. It included 23 sampling locations at which XAD-PAS were deployed for one year and samples were taken approximately every month. The results were evaluated with regard to tempo-spatial variations and sources attribution was done using correlations, Hysplit backward trajectories and a PMF receptor models.**

**As most available studies on PFAAs in the atmosphere are derived from single or only a few sampling sites, this nationwide study is of interest to the international community to better understand the atmospheric distribution of PFAAs. Additionally, China is a country of specific interest as large parts of the PFAS production were shifted from countries in Western Europe, the US and Japan to China and other Asian countries.**

==Response:== Thanks for your appraisal for ou manuscript. We appreciate your valuable comments for improving our manuscript.

**Major comment:**

==Query 1.== **A major query refers to the description and discussion of the used sampling technique. In different parts of the manuscript, it is stated that XAD-PAS collects representative portions of both the particle and the gas phase (line 62, line 191). However, it is reported in other publications that XAD-PAS collects primarily the gas phase (Lai et al., 2018; Melymuk et al., 2014). This difference should be discussed somewhere in the manuscript.**

==Response:== Thanks for the reviewer's good suggestion. Due to the XAD-PAS sampler design,the atmospheric aerosols bound PFAAs could moved into the sampler. Especially for the aerosol size distributions of particle bound PFASs varied with individual specie, e.g. the airborne PFASs, PFOA was predominantly (>70%) observed in small size fraction (<0.14 μm) (Dreyer et al., 2015). In fact, Okeme et al., (2016) employed XAD-Pocket PAS to sample gaseous and particulate SVOCs in indoor environment, with finding not consistent with previous result that

XAD-PAS could sample gas phase pollutant singly. And suggested that the sample efficiency of XAD sorbent sampler for gaseous and particulate phases pollutants need further investigations.

Considering reviewer's suggestion, this differences were discussed and added, detailed as following:

(1). The description of "However, recent field studies have confirmed their occurrence in gaseous phase (Lai et al., 2018;Cassandra et al., 2018;Ahrens et al., 2013)." in lines 57-58 in the original manuscript, was changed to "However, recent field studies have confirmed their occurrence in gaseous phase (Cassandra et al., 2018;Ahrens et al., 2013), e.g. Fang et al., (2018) found the total concentrations of C2, C4–C10 PFCAs and C6 and C8 PFSAs in the gas phase were 0.076–4.0 pg/m$^3$ in the air above the Bohai and Yellow Seas, China." in lines 61-64 in the revised manuscript.

(2). The description of "Fortunately, a number of reports showed that the XAD (a styrene–divinylbenzene copolymer) impregnated sorbent based passive air sampler (SIP–PAS) and XAD based PAS (XAD–PAS), were proven to be an ideal alternative sampling tool for monitoring PFAAs in a wide region, which was suggested to collect a representative sample of both gas and particle phases (Lai et al., 2018;Pavlína et al., 2018)." in lines 60-63 in the original manuscript, was changed to "Fortunately, a number of reports showed that the XAD (a styrene–divinylbenzene copolymer) impregnated sorbent based passive air sampler (SIP–PAS) and XAD based PAS (XAD–PAS), were proven to be an ideal alternative sampling tool for monitoring PFAAs in a wide region. Despite several publications suggested XAD-PAS collects primarily gaseous PFAAs (Melymuk et al., 2014; Lai et al., 2018) in the ambient, current findings were not consistent. Due to the unimpeded movements of particles into the sampler, XAD–PAS was indicated to collect a representative sample of both gas and particle phases (Ahrens et al., 2013; Okeme et al., 2016; Karásková et al., 2018). Moreover, the dominant sorbent for fluorinated compounds was reported as XAD resin in the XAD impregnated SIP–PAS, instead of PUF themselves (Krogseth et al., 2013)." in lines 65-73 in the revised manuscript.

(3). We added the description of "The particle size of XAD-2 is ~20-60 mesh, with water content of 20%-45%, its specific surface area ≥430 m$^2$/g, and the reference adsorption capacity ≥35 mg/g. We should keep in mind that the unimpeded movement of particle bound PFAAs would be captured during sampling using XAD-PAS, which cannot differentiate PFAAs between gas and particle phases. Despite some research suggest the sampling efficiency of gas and particle phase PFAAs were similar (Karásková et al., 2018). In the present study, the two phases PFAAs sampled by XAD-PAS were treated as the whole atmosphere PFAAs concentration." in lines 106-111 in the revised manuscript.

**Query 2.** Moreover, the comparison of the reported concentrations with measurements in other regions in section 3.1 can be skewed because of different sampling techniques and sampling media. If a comparison like this is done, the differences between the sampling techniques and their possible effects on the results should be discussed in a paragraph.

*Response:* According to reviewer's suggestion, the limitation of direct comparison between PFAAs concentration and other measurements was discussed and added in the revised manuscript, as following: "Although there existed inherent differences of PFAAs levels between regions, the impacts from differences in sampling techniques and sorbents between XAD-PAS and SIP-PAS could not be neglected. As indicated by previous researches, XAD has much higher sorptive capacity of PFASs than PUF, wind speed and temperature displayed different degrees of impact on their sampling capacity among different regions. Additionally, UV radiation has the potential to degrade PFAAs due to $O_3$, OH·, and other atmospheric oxidants during sampling. " in lines 203-207 in the revised manuscript.

**Query 3.** The manuscript is well structured and the reader can easily follow the drain of thoughts. However, it still contains several typing and grammar errors. Some are addressed in the section "technical corrections", but this is not exhaustive. Further proofreading by a native speaker would improve the manuscript.

*Response:* Thanks for the reviewer's careful reading on reviewing our manuscript. According to reviewer's suggestion, we have sent the revised manuscript to a professional English language editing service provider in science. This revised manuscript was revised carefully and checked line by line, numerous grammatical mistakes and errors were corrected. For example, "to investigate their occurences" in line 12 in the original manuscript, was reworded as "to investigate their occurrences" in line 13 in the revised manuscript; "was reported ranks as" in line 167 in the original manuscript, was changed to "was reported to rank as" in line 188 in the revised manuscript.

**Query 4.** Specific comments - The number of significant digits should be consistent throughout the manuscript.

*Response:* According to reviewer's suggestion, the number of significant digits of concentrations were revised, and kept consistent throughout the manuscript.

**Introduction**

**Query 5.** -Line 24: PFASs include per- and polyfluoroalkyl substances and not only polyfluoroalkyl substances as stated in this line.

*Response:* As suggested by reviewer, the description of line 24 in the original manuscript was changed to "Perfluoroalkyl acids (PFAAs) are one class of ionic polyfluoroalkyl substances (PFASs), which have excellent characteristics in terms of chemical and thermal stability, high surface activity, and water and oil repulsion (Lindstrom et al., 2011;Wang et al., 2014)." in line 26 in the revised manuscript.

**Query 6.** -Line 32: In the PFAS community, usually the definition of Buck et al. (2011) is used to differentiate between short- and long-chain homologues. According to this, long-chain PFCAs possess 8 or more carbon atoms (7 perfluorinated carbon atoms plus the carboxy group).

*Response:* According to reviewer's suggestion, the classification of long-chain and short-chain PFAAs homologues were revised based on study of Buck et al. (2011). The description of "Of the PFAAs, the long–chain (C ≥7) perfluoroalkyl carboxylic acids (PFCAs) and (C ≥6) perfluoroalkyl sulfonic acids (PFSAs) are more toxic and bio–accumulative than their short–chain analogues (Konstantinos et al., 2010)." in line 32 in the original manuscript, was reworded as "Of the PFAAs, the long–chain (C ≥8) perfluoroalkyl carboxylic acids (PFCAs) and (C ≥7) perfluoroalkyl sulfonic acids (PFSAs) are more toxic and bio–accumulative than their short–chain analogues (Buck et al., 2011)." in lines 35-36 in the revised manuscript.

Also, the corresponding result was revised, e.g. "To the contrary, a recent measurement found the long chain (C ≥ 8) PFCAs were much higher which conducted in the landfill atmosphere in Tianjin, China (Tian et al., 2018)." in lines 159-161 in the original manuscript, was changed to "Similarly, a recent PFAAs measurement conducted in the landfill atmosphere in Tianjin, China (Tian et al., 2018), found the long chain PFCAs were much higher than the short species." in lines 183-184 in the revised manuscript.

**Query 7.** -Line 35/36: In May this year, the Parties to the Stockholm Convention adopted the listing of PFOA to Annex A. It would be good to add this new development to the text.

*Response:* According to reviewer's suggestion, the description of "This especially applies to perfluorooctanoic acid (PFOA) and perfluorohexane sulfonate (PFHxS) for which have been regulated in numerous countries, while perfluorooctane sulfonate (PFOS) have been added to Annex B of the Stockholm Convention in 2009 (Johansson et al., 2008)." in lines 34-36 in the original manuscript, was reworded as "This especially applies to perfluorooctanoic acid (PFOA), perfluorooctane sulfonate (PFOS) and perfluorohexane sulfonate (PFHxS), in which PFOS and PFOA have been added to Annex B and Annex A of the Stockholm Convention in 2009 and 2019, respectively, while PFHxS was under review by the Persistent Organic Pollutants Review Committee (Johansson et al., 2008; UNEP Stockholm Convention, 2019)." in lines 36-40 in the revised manuscript.

**Query 8. -Line 63: Your references "Pavlina K et al., 2018" and "Karaskova P et al., 2018", used later in the manuscript, is in fact the same publication. Please change it to "Karaskova P et al, 2018" in the whole manuscript, as Karaskova (not Pavlina) is the family name of the author.**

*Response:* Thanks for the reviewer's hard work on reviewing our manuscript. The reference of "Pavlina K et al., 2018" was changed to "Karaskova P et al., 2018" in the revised manuscript, and the reference of "Pavlina K et al., 2018" was deleted in the revised manuscript.

**Material and Methods**

**Query 9. -Lines 85-86: Please add the number of sampling sites for each of the seven divisions.**

*Response:* Considering reviewer's suggestion, the description of "These sampling sites were divided into seven administrative divisions: norther of China (NC), southern of 86 China (SC), central of China (CC), eastern of China (EC), northwest of China (NW), northeast of China (NE), and southwest of China (SW)." in lines 85-87 in the original manuscript, was reworded as "These sampling sites were divided into seven administrative divisions: norther China (NC, n=3 sites), southern China (SC, n=2), central China (CC, n=3), eastern China (EC, n=7), northwest of China (NW, n=3), northeast of China (NE, n=2), and southwest of China (SW, n=3)." in lines 97-100 in the revised manuscript.

**Query 10. -Line 87: It would be helpful for the reader to understand from Figure S1 which sampling site belongs to which region (NC, SC etc.). This information could be given in the map itself or in the figure caption.**

*Response:* Considering reviewer's suggestion, the information of each sampling site belonging to which region was added in Figure S1 the revised manuscript. Detailed revision was as following:

[Figure]

[Figure]

**Figure R3.** Revised figure S1 in the manuscript, the upper for the original one, the bottom for the revised figure.

**Query 11.** -Line 121: Usually, "A" refers to the aqueous phase and "B" to the organic solvent, not the other way round. It would avoid misunderstandings if this was turned around.

*Response:* Considering reviewer's suggestion, the description of "The gradient elution program of the mobile phase A (methanol) and B (5 mmol/L aqueous ammonium acetate) was 20% A + 80% B at the start, 95% A + 5% B at 8 min, 100% a at 13 min, 20% A + 80% B at 14 min, and was maintained for 6 min." in lines 120-122 in the original manuscript, was reworded as "The gradient elution program of the mobile phase A (5 mmol/L aqueous ammonium acetate) and B (methanol) was 80% A + 20% B at the start, 5% A + 95% B at 8 min, 100% a at 13 min, 80% A + 20% B at 14 min, and was maintained for 6 min." in lines 139-141 in the revised manuscript.

**Query 12.** -Line 126: There should be a reference to Table S3, which includes the mass transitions.

*Response:* Considering reviewer's suggestion, references of " Karásková et al., 2018" and "Liu et al., 2015" were added to this table in the revised manuscript.

**Query 13.** -Line 134: Do the results refer to the linear isomer, e.g. of PFOS, or to the sum of all isomers?

**Response:** Thanks for reviewer's good suggestion, this result refer to the liner isomer.

**Query 14.** -Line 138: Please add the information, which PFAAs could be detected in which type of blanks and with which standard deviations, either in the text or in Table S3.

**Response:** Considering reviewer's suggestion, the detailed information of filed blanks and laboratory blanks was added in the revised Table S3 in the revised manuscript, see the following Table R2 (same as Table R1).

**Table R2 (Table S3).** MS parameters, MDLs, LODs, LOQs values, recovery rates and blank values for individual compounds of PFAAs

| Analogues | Parent ions (m/z) | Daughter ion (m/z) | Declustering potential (V) [a] | Collision energy (eV) [b] | Retention time (s) | MDLs (pg/m³) | LODs (pg/m³) | LOQs (pg/m³) | Recovery rate (%) | Filed bank (pg/m³) | Laboratory blank (pg/m³) | Internal Standards |
|---|---|---|---|---|---|---|---|---|---|---|---|---|
| PFCAs | | | | | | | | | | | | |
| PFPeA | 263 | 219 | -40 | −34 | 3.16 | 0.41 | 0.31 | 1.05 | 96±17 | 0.41±0.14 | 0.22±0.17 | 1,2–$^{13}C_2$–PFHxA |
| PFHxA | 313 | 269 | -35 | −36 | 3.42 | 0.18 | 0.14 | 0.47 | 108±22 | 0.48±0.06 | 0.37±0.39 | 1,2–$^{13}C_2$–PFHxA |
| PFHpA | 363 | 319→169 | -55 | −28 | 3.70 | 0.22 | 0.16 | 0.55 | 93±16 | 0.62±0.07 | 0.22±0.32 | 1,2,3,4–$^{13}C_4$–PFOA |
| PFOA | 413 | 369→169 | -45 | −39 | 3.99 | 0.33 | 0.26 | 0.87 | 91±13 | 0.93±0.11 | 0.41±0.29 | 1,2,3,4–$^{13}C_4$–PFOA |
| PFNA | 463 | 419→219 | -40 | −44 | 4.32 | 0.61 | 0.46 | 1.53 | 89±17 | 0.57±0.20 | 0.20±0.25 | 1,2,3,4,5–$^{13}C_5$–PFNA |
| PFDA | 513 | 469→219 | -50 | −47 | 4.67 | 0.56 | 0.42 | 1.39 | 93±11 | 0.35±0.19 | 0.28±0.22 | 1,2–$^{13}C_2$–PFDA |
| PFUdA | 563 | 519→269 | -45 | −61 | 5.02 | 0.28 | 0.21 | 0.70 | 88±16 | 0.31±0.09 | 0.31±0.13 | 1,2–$^{13}C_2$–PFUdA |

| | | | | | | | | | | | | |
|---|---|---|---|---|---|---|---|---|---|---|---|---|
| PFDoA | 613 | 569→169 | -45 | −65 | 5.35 | 0.28 | 0.21 | 0.70 | 94±18 | 0.44±0.09 | 0.15±0.18 | 1,2–$^{13}$C$_2$–PFDoA |
| PFTrDA | 663 | 619→169 | -50 | −59 | 5.64 | 0.34 | 0.26 | 0.87 | 102±17 | 0.09±0.11 | 0.05±0.11 | 1,2–$^{13}$C$_2$–PFDoA |
| PFTeDA | 713 | 669→169 | -65 | −57 | 5.94 | 0.14 | 0.31 | 1.03 | 97±21 | 0.12±0.14 | 0.06±0.13 | 1,2–$^{13}$C$_2$–PFDoA |
| **PFSAs** | | | | | | | | | | | | |
| PFBS | 299 | 80→99 | -45 | −64 | 3.19 | 0.25 | 0.20 | 0.66 | 81±25 | 0.11±0.08 | 0.27±0.46 | $^{18}$O$_2$–PFHxS |
| PFHxS | 399 | 80→99 | -55 | −87 | 3.70 | 0.16 | 0.12 | 0.40 | 86±13 | 0.16±0.05 | 0.42±0.27 | $^{18}$O$_2$–PFHxS |
| PFOS | 499 | 80→99 | -55 | −98 | 4.31 | 0.24 | 0.19 | 0.63 | 95±15 | 0.75±0.08 | 0.54±0.61 | 1,2,3,4–$^{13}$C$_4$–PFOS |
| **Internal Standards** | | | | | | | | | | | | |
| 1,2–$^{13}$C$_2$–PFHxA | 315 | 270 | -75 | −41 | 3.40 | / | / | / | / | / | / | / |
| 1,2,3,4–$^{13}$C$_4$–PFOA | 417 | 372 | -40 | −41 | 3.99 | / | / | / | / | / | / | / |
| 1,2,3,4,5–$^{13}$C$_5$–PFNA | 468 | 423 | -84 | −52 | 4.34 | / | / | / | / | / | / | / |
| 1,2–$^{13}$C$_2$–PFDA | 515 | 470 | -87 | −51 | 4.69 | / | / | / | / | / | / | / |
| 1,2–$^{13}$C$_2$–PFUdA | 565 | 520 | -79 | −61 | 5.02 | / | / | / | / | / | / | / |
| 1,2–$^{13}$C$_2$–PFDoA | 615 | 570 | -66 | −55 | 5.35 | / | / | / | / | / | / | / |
| $^{18}$O$_2$–PFHxS | 403 | 103 | -55 | 97 | 3.72 | / | / | / | / | / | / | / |
| 1,2,3,4–$^{13}$C$_4$–PFOS | 503 | 80 | -80 | 97 | 4.31 | / | / | / | / | / | / | / |

[a]: cited from Karásková et al., 2018.

[b]: cited from Karásková et al., 2018 and Liu et al., 2015.

**Results and Discussion**

**Query 15. -** **How are results below MDL given in this table? Does "0" refer to values below MDL? Please include this information.**

*Response:* Thank very much for reviewer's suggestion. The measured abundances of PFAAs which below MDL was marked as "0" in the original manuscript. Considering reviewer's suggestion and some statistics standard used, we have modified these values to "BDL" in Table S4 in the revised Supporting Materials, as following:

**Table R3(Table S4).** The measured abundances of PFAAs in this study(n=268)

| Analogues | Detection frequency (%) | Average value (pg/m$^3$) | Standard deviation (pg/m$^3$) | Minimum value (pg/m$^3$) | Maximum value (pg/m$^3$) | Median value (pg/m$^3$) |
|---|---|---|---|---|---|---|
| PFCAs | | | | | | |
| PFPeA | 84.8 | 4.96 | 4.77 | BDL | 35.2 | 3.55 |
| PFHxA | 92.1 | 5.36 | 7.17 | BDL | 79.7 | 3.73 |
| PFHpA | 94.7 | 3.42 | 3.71 | BDL | 28.9 | 2.39 |
| PFOA | 100 | 8.19 | 8.03 | 0.36 | 70.4 | 6.24 |
| PFNA | 96.6 | 3.07 | 2.77 | BDL | 22.7 | 2.52 |
| PFDA | 96.2 | 4.13 | 3.74 | BDL | 30.5 | 3.36 |
| PFUdA | 75.6 | 1.24 | 1.32 | BDL | 6.72 | 0.86 |
| PFDoA | 63.5 | 0.56 | 0.50 | BDL | 3.18 | 0.45 |
| PFTrDA | 37.3 | 0.58 | 0.56 | BDL | 3.57 | 0.47 |
| PFTeDA | 41.7 | 0.19 | 0.25 | BDL | 2.25 | 0.11 |
| PFSAs | | | | | | |
| PFBS | 62.2 | 1.96 | 1.85 | BDL | 9.39 | 1.37 |
| PFHxS | 71.6 | 0.99 | 1.38 | BDL | 13.2 | 0.56 |
| PFOS | 100 | 5.20 | 4.30 | 0.34 | 25.5 | 3.87 |

BDL: below detection limit.

**Query 16. -** **For some of the results, the median value is below the MDL given in Table S3 (e.g. for PFTeDA).**

**How did you calculate these median values?**

*Response:* Thanks for the reviewer's suggestion. We value this suggestion very highly, checked these results carefully and found the MDL of PFTeDA was 0.14 rather 0.41, and revised this mistake. For the statistics analysis of measured concentrations, the results of BDL were replaced by 1/2 of the corresponding MDL values.

Considering reviewer's suggestion, the description of "Statistical analyses were carried out by SPSS Statistics 22 (IBM Inc. US) and SigmaPlot 14.0 (Systat Software, US)." in line 147 in the original manuscript, was changed to "Statistical analyses were carried out by SPSS Statistics 22 (IBM Inc. US), and the values of 1/2 MDL were used to replace these measured results of BDL. The statistics figures were depicted using technical software of SigmaPlot 14.0 (Systat Software, US)." in lines 167-169 in this revised manuscript.

**Query 17.** -Table 1: It would be helpful to know, which "PFAAs" are included in the sum given in the fifth column.

*Response:* Considering reviewer's suggestion, more detailed information on species PFAAs was added in the note of "b"of PFAAs in Table 1, it was reworded as "b: represent the total concentration ranges of PFCAs and PFSAs; mean concentrations of the total PFCAs and PFSAs;" in Table 1 in the revised manuscript.

**Query 18.** -Line 199: It would be interesting which type of manufacturers are included in figure S3 and which industries are not?

*Response:* Considering reviewer's suggestion, more detailed information on the fluoride manufacturer was added to the caption of Figure S3 in the revised Supporting Materials, "Figure S3. The spatial distributions of fluoride related products manufacturers in China (note that part of fluoride related industries were not included in this figure) and the different geographical conditions" "Figure S3. The spatial distributions of fluoride related products manufacturers in China and the different geographical conditions (note that the fluoride related manufacturers including textiles, crude plastic, paint coating, packaging materials, while part of fluoride related industries were not included in this figure)"

**Query 19.** -Line 201 to 209: Was this monthly variation stronger for specific sampling sites than for others?

*Response:* The monthly variations of PFAAs varied based on site environments and local geographical conditions, the monthly variations of PFAAs in each site differed more or less. As shown in Figure S2, Beijing, Tianjin, and Xinjiang sites shared a similar monthly PFAAs variations, while Shanghai displayed a much different trend, which was controlled by local sources emissions as well.

**Query 20.** -Line 272 to 274: It would be helpful for the reader to get a short explanation (1-2 sentences) why the air mass origins shown in figure S5 were a driving factor for PFAA variation.

*Response:* Considering reviewer's suggestion, an explanation of the air mass origins in Table S5 was added in the manuscript, as "As illustrated in Figure S5, the 48 hours back trajectories were generally associated with air masses originating from the surrounding areas of the sampling locations, the trajectories which overlapped with urban areas in Zhejiang, Jiangxi and Shanghai, which confirmed that the air mass origins was a driving factor for PFAAs variation. " in lines 299-302 in the revised manuscript.

**Query 21.** -Line 300: The production of PFOA to use it as emulsifier in PTFE manufacturing is also an important direct source in China, isn't it?

*Response:* We strongly agree with reviewer's suggestion, since PFOA is widely used in the manufacturing of polytetrafluoroethylene (PTFE), perfluorinated ethylene propolymer (FEP), and perfluoroalkoxy polymers (PFA). Considering reviewer's suggestion, the description of "PFOA was considered as the marker for the emulsification of plastics, rubber products, flame retardants for textiles, paper surface treatments, and fire foams (Liu et al., 2015;Konstantinos et al., 2010)." in lines 299-300 in the original manuscript, was reworded as "PFOA was considered as the marker for the emulsification of plastics, rubber products, flame retardants for textiles, paper surface treatments, fire foams and PTFE emulsifiers (Liu et al., 2015;Konstantinos et al., 2010)." in lines 327-328 in the revised manuscript.

**Query 22.** -Line 331: You state in the conclusion that the measured PFAAs were "several times to several magnitudes higher" than other urban atmosphere levels. This is not that obvious when reading 3.1 and looking at table 1. For example, the values reported for Brno are in a similar range as the results from this study, if I understand it correctly?

*Response:* Considering reviewer's suggestion, this description of this conclusion was modified. "Results indicated that the measured PFAAs were several times to several magnitudes higher than other urban atmosphere levels, and much higher abundances existed in winter seasons compared with in the summer." in lines 330-332 in the original manuscript, was changed to "Results indicated that the measured PFAAs in the present study were several times to several magnitudes higher than the levels conducted in most other urban locations, while far lower than the measurements implemented at point sources. In which, the C5–C14 PFCAs analogues occupied 79.6% of the total PFAAs variations, PFOA, PFHxA and PFOS ranked the top three species. Additionally," in lines 358-362 in the revised manuscript.

**Technical corrections**

**Query 23.** **- Line 15/16 "perfluorohexanoic" and "perfluoroheptanoic" have to be without "-"**

*Response:* According to reviewer's suggestion, the "perfluoro–hexanoic acid (PFHxA)", and "perfluoro–heptanoic acid (PFPeA)" was revised as "perfluoro–hexanoic acid (PFHxA)" and "perfluoro–heptanoic acid (PFPeA)" in the revised manuscript, respectively.

**Query 24.** **-Line 21: It has to be "fluorotelomer-based" instead of "fluoro-telomere based"**

*Response:* Considering reviewer's suggestion, "fluoro-telomere based" in line 21 in the original manuscript was reworded as "fluorotelomer-based" in line 22 in the revised manuscript.

**Query 25.** **- Line 65: "deployed" instead of "depolyed"**

*Response:* Considering reviewer's suggestion, "depolyed" in line 65 in the original manuscript was reworded as "deployed" in line 74 in the revised manuscript.

**Query 26.** **-Lines 85-86. I think it has to be "north of China (NC)" or "northern China (NC)" instead of "northern of China (NC)". This also applies to the other regions.**

*Response:* Considering reviewer's suggestion, these related description were modified. For example, in the section of "2.2 Sample Collection", "These sampling sites were divided into seven administrative divisions: norther of China (NC), southern of China (SC), central of China (CC), eastern of China (EC), northwest of China (NW), northeast of China (NE), and southwest of China (SW)." in line 85-87 in the original manuscript, was changed to "These sampling sites were divided into seven administrative divisions: norther China (NC, n=3 sites), southern China (SC, n=2), central China (CC, n=3), eastern China (EC, n=7), northwest of China (NW, n=3), northeast of China (NE, n=2), and southwest of China (SW, n=3)." in lines 98-100 in the revised manuscript.

In abstract, the description of "Spatially, the content of PFAAs displayed a declining gradient trend of central areas > eastern areas > western areas, and Henan contributed as the largest proportion of PFAAs." in lines 17-19 in the original manuscript, was changed to "Spatially, the content of PFAAs displayed a declining gradient trend of central China> northern China> eastern China> northeast of China> southwest of China> northwest of China> southern China areas, and Henan contributed as the largest proportion of PFAAs." in lines 18-20 in the revised manuscript.

**Query 27.** -Line 160: "which conducted in the landfill atmosphere in Tianjin" does not connect to the rest of the sentence.

*Response:* Considering reviewer's suggestion, "To the contrary, a recent measurement found the long chain (C ≥ 8) PFCAs were much higher which conducted in the landfill atmosphere in Tianjin, China (Tian et al., 2018)." in lines 159-161 in the original manuscript, was changed to "Similarly, a recent PFAAs measurement conducted in the landfill atmosphere in Tianjin, China (Tian et al., 2018), found the long chain PFCAs were much higher than the short species." in lines 183-184 in the revised manuscript.

**Query 28.** -Line 167: "neutral PFASs in Chinese air" instead of "neural PFASs in China air"

*Response:* Considering reviewer's suggestion, "Meanwhile, one major variation of PFOA precursor, 8:2 FTOH, was reported ranks as the highest concentration among neural PFASs in China air" in lines 166-167 in the original manuscript, was changed to "Meanwhile, one major variation of PFOA precursor, 8:2 FTOH, was reported to rank as the highest concentration among neural PFASs in air of China" in lines 190-191 in the revised manuscript.

**Query 29.** -Line 189: "may be could attribute" is ungrammatical.

*Response:* Considering reviewer's suggestion, "The winter maxima abundance of PFAAs may be could attribute to the stagnant atmospheric conditions," in lines 189-190 in the original manuscript, was changed to "The winter maxima abundance of PFAAs could be attribute to the stagnant atmospheric conditions," in lines 216-217 in the revised manuscript.

**Query 30.** -Line 318: correlations "to" each other

*Response:* Considering reviewer's suggestion, the description of "In addition, these four analogues showed apparent positive correlations each other (r =0.59–0.79, p<0.01)." in lines 317-318 in the original manuscript, was changed to "In addition, these four analogues showed apparent positive correlations to each other (r =0.59–0.79, p<0.01)." in lines 345-346 in the revised manuscript.

We tried our best to improve the manuscript and made some changes in the manuscript. These changes will not influence the content and framework of the paper. We appreciate for Editors/ Reviewers' warm work earnestly, and hope that the correction will meet with approval. Once again, thanks very much for your comments and suggestions.

Yours sincerely,

Best regards!

Deming Han

Tel: +86 21 54743936

Fax: (86 21) 5474 0825

E-mail: handeem@sjtu.edu.cn

Add.:800 Dongchuan Road, Minhang District Shanghai, China

[revised manuscript text omitted]